# Gaussian Differential Privacy on Riemannian Manifolds

**Yangdi Jiang, Xiaotian Chang, Yi Liu, Lei Ding, Linglong Kong, Bei Jiang\***
Department of Mathematical and Statistical Sciences
University of Alberta
`{yangdi, xchang4, yliu16, lding1, lkong, bei1}@ualberta.ca`

## Abstract

We develop an advanced approach for extending Gaussian Differential Privacy (GDP) to general Riemannian manifolds. The concept of GDP stands out as a prominent privacy definition that strongly warrants extension to manifold settings, due to its central limit properties. By harnessing the power of the renowned Bishop-Gromov theorem in geometric analysis, we propose a Riemannian Gaussian distribution that integrates the Riemannian distance, allowing us to achieve GDP in Riemannian manifolds with bounded Ricci curvature. To the best of our knowledge, this work marks the first instance of extending the GDP framework to accommodate general Riemannian manifolds, encompassing curved spaces, and circumventing the reliance on tangent space summaries. We provide a simple algorithm to evaluate the privacy budget $\mu$ on any one-dimensional manifold and introduce a versatile Markov Chain Monte Carlo (MCMC)-based algorithm to calculate $\mu$ on any Riemannian manifold with constant curvature. Through simulations on one of the most prevalent manifolds in statistics, the unit sphere $S^d$, we demonstrate the superior utility of our Riemannian Gaussian mechanism in comparison to the previously proposed Riemannian Laplace mechanism for implementing GDP.

## 1 Introduction

As technological advancements continue to accelerate, we are faced with the challenge of managing and understanding increasingly complex data. This data often resides in nonlinear manifolds, commonly found in various domains such as medical imaging [Pennec et al., 2019, Dryden, 2005, Dryden et al., 2009], signal processing [Barachant et al., 2010, Zanini et al., 2018], computer vision [Turaga and Srivastava, 2015, Turaga et al., 2008, Cheng and Vemuri, 2013], and geometric deep learning [Belkin et al., 2006, Niyogi, 2013]. These nonlinear manifolds are characterized by their distinct geometric properties, which can be utilized to extract valuable insights from the data.

As data complexity grows, so does the imperative to safeguard data privacy. Differential Privacy (DP) [Dwork et al., 2006b] has gained recognition as a prominent mathematical framework for quantifying privacy protection, and a number of privacy mechanisms [McSherry and Talwar, 2007, Barak et al., 2007, Wasserman and Zhou, 2010, Reimherr and Awan, 2019] have been devised with the aim of achieving DP. Conventional privacy mechanisms, while effective for dealing with linear data, encounter difficulties when handling complex non-linear data. In such cases, a common approach, referred to as the extrinsic approach, is to embed the non-linear data into the ambient Euclidean space, followed by the application of standard DP mechanisms. However, as exemplified in the work of Reimherr et al. [2021], the intrinsic properties of such non-linear data enable us to achieve better data utility while simultaneously preserving data privacy. Therefore, it is imperative that privacy mechanisms adapt to the complexity of non-linear data by employing tools from differential geometry to leverage the geometric structure within the data.

37th Conference on Neural Information Processing Systems (NeurIPS 2023).

**Related Work**   Reimherr et al. [2021] is the first to consider the general manifolds in the DP litera-ture. It extends the Laplace mechanism for $\varepsilon$-DP from Euclidean spaces to Riemannian manifolds. Focusing on the task of privatizing Frechet mean, it demonstrates that better utility can be achieved when utilizing the underlying geometric structure within the data. Continuing its work, Soto et al. [2022] develops a K-norm gradient mechanism for $\varepsilon$-DP on Riemannian manifolds and shows that it outperforms the Laplace mechanism previously mentioned in the task of privatizing Frechet mean. Similarly, Utpala et al. [2023b] extends $(\varepsilon, \delta)$-DP and its Gaussian mechanism but only to one specific manifold, the space of symmetric positive definite matrices (SPDM). Equipping the space of SPDM with the log Euclidean metric, it becomes a geometrically flat space [Arsigny et al., 2007]. This allows them to simplify their approach and work with fewer complications, although at the expense of generality. In contrast to the task of releasing manifold-valued private summary, Han et al. [2022], Utpala et al. [2023a] focus on solving empirical risk minimization problems in a $(\varepsilon, \delta)$-DP compliant manner by privatizing the gradient which resides on the tangent bundle of Riemannian manifolds. Working on tangent spaces instead of the manifold itself, they could bypass many of the difficulties associated with working under Riemannian manifolds.

**Motivations**   Although the $\varepsilon$-differential privacy (DP) and its Laplace mechanism have been ex-tended to general Riemannian manifolds in Reimherr et al. [2021], there are other variants of DP [Dwork et al., 2006a, Mironov, 2017, Bun and Steinke, 2016, Dong et al., 2022]. Each of them possesses unique advantages over the pure DP definition, and therefore their extensions should be considered as well. As one such variant, GDP offers superior composition and subsampling properties to that of $\varepsilon$-DP. Additionally, it's shown that all hypothesis testing-based privacy definitions converge to the guarantees of GDP in the limit of composition [Dong et al., 2022]. Furthermore, when the dimension of the privatized data approaches infinity, a large class of noise addition private mechanisms is shown to be asymptotically GDP [Dong et al., 2021]. These traits establish GDP as the focal privacy definition among different variants of DP definitions and therefore make it the most suitable option for generalizing to Riemannian manifolds.

**Main Contributions**   With the goal of releasing manifold-valued statistical summary in a GDP-compliant manner, we extend the GDP framework to general Riemannian manifolds, establishing the ability to use Riemannian Gaussian distribution for achieving GDP on Riemannian manifolds. We then develop an analytical form to achieve $\mu$-GDP that covers all the one-dimensional cases. Furthermore, we propose a general MCMC-based algorithm to evaluate the privacy budget $\mu$ on Riemannian manifolds with constant curvature. Lastly, we conduct numerical experiments to evaluate the utility of our Riemannian Gaussian mechanism by comparing it to the Riemannian Laplace mechanism. Our results conclusively demonstrate that to achieve GDP, our Gaussian mechanism exhibits superior utility compared to the Laplace mechanism.

## 2   Notation and Background

In this section, we first cover some basic concepts from Riemannian geometry. The materials covered can be found in standard Riemannian geometry texts such as Lee [2006], Petersen [2006], Pennec et al. [2019], Said [2021]. Then we review some definitions and results on DP and GDP, please refer to Dwork and Roth [2014], Dong et al. [2022, 2021] for more detail.

### 2.1   Riemannian Geometry

Throughout this paper we let $\mathcal{M}$ denote a $d$-dimensional complete Riemannian manifold unless stated otherwise. A Riemannian metric $g$ is a collection of scalar products $\langle \cdot, \cdot \rangle_x$ on each tangent space $T_x\mathcal{M}$ at points $x$ of the manifold that varies smoothly from point to point. For each $x$, each such scalar product is a positive definite bilinear map $\langle \cdot, \cdot \rangle_x : T_x\mathcal{M} \times T_x\mathcal{M} \to \mathbb{R}$.

Equipped with a Riemannian metric $g$, it grants us the ability to define length and distance on $\mathcal{M}$. Consider a curve $\gamma(t)$ on $\mathcal{M}$, the length of the curve is given by the integral

$$L(\gamma) = \int \|\dot{\gamma}(t)\|_{\gamma(t)} dt = \int \left( \langle \dot{\gamma}(t), \dot{\gamma}(t) \rangle_{\gamma(t)} \right)^{\frac{1}{2}} dt$$

where the $\dot{\gamma}(t)$ is the velocity vector and norm $\|\dot{\gamma}(t)\|$ is uniquely determined by the Riemannian metric $g$. Note we use $\| \cdot \|$ to denote the $l_2$ norm throughout this paper.

It follows that the distance between two points $x, y \in \mathcal{M}$ is the infimum of the lengths of all piece-wise smooth curves from $x$ to $y$, $d(x, y) = \inf_{\gamma:\gamma(0)=x,\gamma(1)=y} L(\gamma)$. In a similar fashion, we can introduce the notion of measure on $\mathcal{M}$. The Riemannian metric $g$ induces a unique measure $\nu$ on the Borel $\sigma$-algebra of $\mathcal{M}$ such that in any chart $U$, $d\nu = \sqrt{\det g}d\lambda$ where $g = (g_{ij})$ is the matrix of the Riemannian metric g in $U$, and $\lambda$ is the Lebesgue measure in $U$ [Grigoryan, 2009].

Given a point $p \in \mathcal{M}$ and a tangent vector $v \in T_p\mathcal{M}$, there exists a unique geodesic $\gamma_{(p,v)}(t)$ starting from $p = \gamma_{(p,v)}(0)$ with tangent vector $v = \dot{\gamma}_{(p,v)}(0)$ defined in a small neighborhood of zero. It can then be extended to $\mathbb{R}$ since we assume $\mathcal{M}$ is complete. This enables us to define the exponential map $\exp_p : T_p\mathcal{M} \to \mathcal{M}$ as $\exp_p(v) = \gamma_{(p,v)}(1)$. For any $p \in \mathcal{M}$, there is a neighborhood $V$ of the origin in $T_p\mathcal{M}$ and a neighborhood $U$ of $p$ such that $\exp_p |_V : V \to U$ is a diffeomorphism. Such $U$ is called a normal neighborhood of $p$. Locally, the straight line crossing the origin in $T_p\mathcal{M}$ transforms into a geodesic crossing through $p$ on $\mathcal{M}$ via this map. On the normal neighborhood $U$, the inverse of the exponential map can be defined and is denoted by $\log_p$. The injectivity radius at a point $p \in \mathcal{M}$ is then defined as the maximal radius $R$ such that $B_p(R) \subset \mathcal{M}$ is a normal neighborhood of $p$, and the injectivity radius of $\mathcal{M}$ is given by $\operatorname{inj}_{\mathcal{M}} = \inf\{\operatorname{inj}_{\mathcal{M}}(p), \ p \in \mathcal{M}\}$.

### 2.2 Differential Privacy

We start this section with the definition of $(\varepsilon, \delta)$-DP.

**Definition 2.1** ([Dwork et al., 2006a])**.** *A data-releasing mechanism $M$ is said to be $(\varepsilon, \delta)$-differentially private with $\varepsilon \geq 0, 0 \leq \delta \leq 1$, if for any adjacent datasets, denoted as $\mathcal{D} \simeq \mathcal{D}'$, differing in only one record, we have $\Pr(M(\mathcal{D}) \in A) \leq e^{\varepsilon} \Pr(M(\mathcal{D}') \in A) + \delta$ for any measurable set $A$ in the range of $M$.*

Differential privacy can be interpreted from the lens of statistical hypothesis testing [Wasserman and Zhou, 2010, Kairouz et al., 2017]. Given the outcome of a $(\varepsilon, \delta)$-DP mechanism and a pair of neighboring datasets $\mathcal{D} \simeq \mathcal{D}'$, consider the hypothesis testing problem with $H_0$: The underlying dataset is $\mathcal{D}$ and $H_1$: The underlying dataset is $\mathcal{D}'$. The smaller the $\varepsilon$ and $\delta$ are, the harder this hypothesis testing will be. That is, it will be harder to detect the presence of one individual based on the outcome of the mechanism. More specifically, $(\varepsilon, \delta)$-DP tells us the power (that is, 1 - type II error) of any test at significance level $\alpha \in [0, 1]$ is bounded above by $e^{\varepsilon}\alpha + \delta$. Using this hypothesis testing interpretation, we can extend $(\varepsilon, \delta)$-DP to the notion of Gaussian differential privacy.

Let $M(\mathcal{D}), M(\mathcal{D}')$ denote the distributions of the outcome under $H_0, H_1$ respectively. Let $T(M(\mathcal{D}), M(\mathcal{D}')) : [0, 1] \to [0, 1], \alpha \mapsto T(M(\mathcal{D}), M(\mathcal{D}'))(\alpha)$ denote the optimal tradeoff between type I error and type II error. More specifically, $T(M(\mathcal{D}), M(\mathcal{D}'))(\alpha)$ is the smallest type II error when type I error equals $\alpha$.

**Definition 2.2** ([Dong et al., 2022])**.** *A mechanism $M$ is said to satisfy $\mu$-Gaussian Differential Privacy ($\mu$-GDP) if $T(M(\mathcal{D}), M(\mathcal{D}')) \geq G_{\mu}$ for all neighboring datasets $\mathcal{D} \simeq \mathcal{D}'$ with $G_{\mu} := T(N(0, 1), N(\mu, 1))$.*

Informally, $\mu$-GDP states that it's harder to distinguish $\mathcal{D}$ from $\mathcal{D}'$ than to distinguish between $N(0, 1)$ and $N(\mu, 1)$. Similar to the case of $(\varepsilon, \delta)$-differential privacy, a smaller value of $\mu$ provides stronger privacy guarantees. As a privacy definition, $\mu$-GDP enjoys several unique advantages over the $(\varepsilon, \delta)$-DP definition. Notably, it has a tight composition property that cannot be improved in general. More importantly, a crucial insight in Dong et al. [2022] is that the best way to evaluate the privacy of the composition of many "highly private" mechanisms is through $\mu$-GDP. More specifically, it gives a central limit theorem that states all hypothesis testing-based privacy definitions converge to the guarantees of $\mu$-GDP in the limit of composition. Furthermore, Dong et al. [2021] shows that a large class of noise addition mechanisms is asymptotic $\mu$-GDP when the dimension of the privatized data approaches infinity. These distinct characteristics position $\mu$-GDP as the focal privacy definition among different variants of DP definitions, and we will extend the $\mu$-GDP framework to general Riemannian manifolds in Section 3.

## 3  Gaussian Differential Privacy on General Riemannian Manifolds

Our primary objective in this study is to disclose a $\mathcal{M}$-valued statistical summary while preserving privacy in a GDP-compliant manner. To this end, we first extend the GDP definition to general

Riemannian manifolds. In Definition 2.2, $\mu$-GDP is defined through the optimal trade-off function $T(M(\mathcal{D}), M(\mathcal{D}'))$, which is challenging to work with on Riemannian manifolds. We successfully resolve this difficulty by expressing $\mu$-GDP as an infinite collection of $(\varepsilon, \delta)$-DP (Corollary 1 in Dong et al. [2022]). Since $(\varepsilon, \delta)$-DP is a well-defined notion on any measurable space [Wasserman and Zhou, 2010], it readily extends to any Riemannian manifold equipped with the Borel $\sigma$-algebra. Following this methodology, we define $\mu$-GDP on general Riemannian manifolds as follows.

**Definition 3.1.** *A $\mathcal{M}$-valued data-releasing mechanism $M$ is said to be $\mu$-GDP if it's $(\varepsilon, \delta_\mu(\varepsilon))$-DP for all $\varepsilon \geq 0$, where*

$$\delta_\mu(\varepsilon) := \Phi\left(-\frac{\varepsilon}{\mu} + \frac{\mu}{2}\right) - \mathrm{e}^\varepsilon \Phi\left(-\frac{\varepsilon}{\mu} - \frac{\mu}{2}\right).$$

*and $\Phi$ denotes the cumulative distribution function of the standard normal distribution.*

Similarly, we extend the notion of sensitivity to Riemannian manifolds as well.

**Definition 3.2** (Reimherr et al. [2021]). *A summary $f$ is said to have a **global sensitivity** of $\Delta < \infty$, with respect to $d(\cdot, \cdot)$, if we have $d\left(f(\mathcal{D}), f\left(\mathcal{D}'\right)\right) \leq \Delta$ for any two dataset $\mathcal{D} \simeq \mathcal{D}'$.*

Following the extension of $\mu$-GDP to Riemannian manifolds, it is crucial to develop a private mechanism that is compliant with $\mu$-GDP. Given that the Gaussian distribution satisfies $\mu$-GDP on Euclidean space [Dong et al., 2022], we hypothesize that an analogous extension of the Gaussian distribution into Riemannian manifolds would yield similar adherence to $\mu$-GDP. (e.g., [Reimherr et al., 2021] for extending Laplace distribution to satisfy $\varepsilon$-DP on Riemannian manifolds). We introduce the Riemannian Gaussian distribution in the following definition.

**Definition 3.3** (Section 2.5 in Pennec et al. [2019]). *Let $(\mathcal{M}, g)$ be a Riemannian manifold such that*

$$Z(\eta, \sigma) = \int_{\mathcal{M}} \exp\left\{-\frac{d(y, \eta)^2}{2\sigma^2}\right\} d\nu(y) < \infty.$$

*We define a probability density function w.r.t $d\nu$ as*

$$p_{\eta, \sigma}(y) = \frac{1}{Z(\eta, \sigma)} \exp\left\{-\frac{d(y, \eta)^2}{2\sigma^2}\right\}. \tag{1}$$

*We call this distribution a **Riemannian Gaussian distribution** with footprint $\eta$ and rate $\sigma$ and denote it by $Y \sim N_{\mathcal{M}}(\eta, \sigma^2)$.*

The necessity for $Z(\eta, \sigma)$ to be finite is generally of little concern, as it has been shown to be so in any compact manifolds [Chakraborty and Vemuri, 2019] or any Hadamard manifolds[1]—with lower-bounded sectional curvature [Said, 2021]. The distribution we introduce has been established to maximize entropy given the first two moments [Pennec et al., 2019, Pennec, 2006], and has already found applications in a variety of scenarios [Zhang and Fletcher, 2013, Hauberg, 2018, Said et al., 2017, Cheng and Vemuri, 2013, Zanini et al., 2018, Chakraborty and Vemuri, 2019]. When $\mathcal{M} = \mathbb{R}^d$, the Riemannian Gaussian distribution reduces to the multivariate Gaussian distribution with a mean of $\eta$ and a variance of $\sigma^2 \mathbf{I}$.

However, it is critical to highlight that this extension is not the only possible method for integrating the Gaussian distribution into Riemannian manifolds. Other approaches are available, such as the heat kernel diffusion process articulated by Grigoryan [2009], or the exponential-wrapped Gaussian introduced by Chevallier et al. [2022]. For an in-depth exploration of sampling from the Riemannian Gaussian distribution defined above, we refer readers to Section 4.1.

Furthermore, the subsequent theorem underscores the ability of the Riemannian Gaussian distribution, as defined herein, to meet the requirements of Gaussian Differential Privacy.

**Theorem 3.1.** *Let $\mathcal{M}$ be a Riemannian manifold with lower bounded Ricci curvature and $f$ be a $\mathcal{M}$-valued summary with global sensitivity $\Delta$. The Riemannian Gaussian distribution with footprint $f(\mathcal{D})$ and rate $\sigma$ satisfies $\mu$-GDP for some $\mu > 0$.*

*Proof.* See Appendix A.1. □

---

[1] A Hadamard manifold is a Riemannian manifold that is complete and simply connected and has everywhere non-positive sectional curvature.

While Theorem 3.1 confirms the potential of the Riemannian Gaussian distribution to achieve GDP, it leaves the relationship between the privacy budget ($\mu$) and the rate ($\sigma$) undefined. The subsequence theorem establishes such a connection:

**Theorem 3.2** (Riemannian Gaussian Mechanism). *Let $\mathcal{M}$ be a Riemannian manifold with lower bounded Ricci curvature and $f$ be a $\mathcal{M}$-valued summary with global sensitivity $\Delta$. The Riemannian Gaussian distribution with footprint $f(D)$ and rate $\sigma > 0$ is $\mu$-GDP if and only if $\mu$ satisfies the following condition, $\forall \varepsilon \geq 0$,*

$$\sup_{\mathcal{D} \simeq \mathcal{D}'} \int_A p_{\eta_1, \sigma}(y) \, d\nu(y) - e^\varepsilon \int_A p_{\eta_2, \sigma}(y) \, d\nu(y) \leq \delta_\mu(\varepsilon) \tag{2}$$

*where $A := \{y \in \mathcal{M} : p_{\eta_1, \sigma}(y)/p_{\eta_2, \sigma}(y) \geq e^\varepsilon\}$ and $\eta_1 := f(\mathcal{D}), \eta_2 := f(\mathcal{D}')$.*

*Proof.* See Appendix A.2. $\qquad\square$

Given the rate $\sigma$, Theorem 3.2 provides us a way of computing the privacy budget $\mu$ through the inequality (2). When $\mathcal{M} = \mathbb{R}^d$, the set $A$ enjoys a tractable form, $A = \{y \in \mathbb{R}^d : \langle y - \eta_2, \eta_1 - \eta_2 \rangle \geq \frac{1}{2}(2\sigma^2\varepsilon/\|\eta_1 - \eta_2\| + \|\eta_1 - \eta_2\|)\}$, and the inequality (2) then reduces to,

$$\sup_{D \simeq D'} \Phi\left(-\frac{\sigma\varepsilon}{\|\eta_1 - \eta_2\|} + \frac{\|\eta_1 - \eta_2\|}{2\sigma}\right) - e^\varepsilon \Phi\left(-\frac{\sigma\varepsilon}{\|\eta_1 - \eta_2\|} + \frac{\|\eta_1 - \eta_2\|}{2\sigma}\right) \leq \delta_\mu(\varepsilon)$$

where the equality holds if and only if $\sigma = \Delta/\mu$, which reduces to the Gaussian mechanism on Euclidean spaces in Dong et al. [2022]. It's worth pointing out that on $\mathbb{R}^d$, the Pythagorean theorem allows us to reduce the integrals to one-dimensional integrals resulting in a simple solution for any dimension $d$. Unfortunately, the lack of Pythagorean theorem on non-Euclidean spaces makes it difficult to evaluate the integrals on manifolds of dimensions greater than one. It's known that any smooth, connected one-dimensional manifold is diffeomorphic either to the unit circle $S^1$ or to some interval of $\mathbb{R}$ [Milnor and Weaver, 1997]. Therefore, we encompass all one-dimensional cases by presenting the following result on $S^1$.

**Corollary 3.2.1** (Riemannian Gaussian Mechanism on $S^1$). *Let $f$ be a $S^1$-valued summary with global sensitivity $\Delta$. The Riemannian Gaussian distribution $N_{\mathcal{M}}\left(f(\mathcal{D}), \sigma^2\right)$ is $\mu$-GDP if and only if $\mu$ satisfies the following condition, $\forall \varepsilon \in [0, \pi\Delta/(2\sigma^2)], h(\sigma, \varepsilon, \Delta) \leq \delta_\mu(\varepsilon)$ where*

$$h(\sigma, \varepsilon, \Delta) = \frac{1}{C(\sigma)}\left[\Phi\left(-\frac{\sigma\varepsilon}{\Delta} + \frac{\Delta}{2\sigma}\right) - e^\varepsilon \Phi\left(-\frac{\sigma\varepsilon}{\Delta} - \frac{\Delta}{2\sigma}\right)\right]$$
$$- \frac{1}{C(\sigma)}\left[\Phi\left(\frac{\sigma\varepsilon}{\Delta} + \frac{\Delta}{2\sigma} - \frac{\pi}{\sigma}\right) - e^\varepsilon \Phi\left(\frac{\sigma\varepsilon}{\Delta} - \frac{\Delta}{2\sigma} + \frac{\pi}{\sigma}\mathbf{1}_{\varepsilon \leq \frac{\Delta^2}{2\sigma^2}} - \frac{\pi}{\sigma}\mathbf{1}_{\varepsilon > \frac{\Delta^2}{2\sigma^2}}\right)\right]$$
$$- e^\varepsilon \mathbf{1}_{\varepsilon \leq \frac{\Delta^2}{2\sigma^2}}$$

*with $C(\sigma) = \Phi(\pi/\sigma) - \Phi(-\pi/\sigma)$.*

*Proof.* See Appendix A.3. $\qquad\square$

In summary, Corollary 3.2.1 provides us the analytical form for the integrals in (2). By specifying the rate $\sigma$ and sensitivity $\Delta$, it becomes feasible to compute the privacy budget $\mu$, which is the smallest $\mu$ such that $h(\sigma, \varepsilon, \Delta) \leq \delta_\mu(\varepsilon)$ for all $\varepsilon \in [0, \pi\Delta/(2\sigma^2)]$. The systematic procedure is summarized in Algorithm 1. It is important to emphasize that the result in Corollary 3.2.1 together with the existing Euclidean space result covers all one-dimensional cases, and for manifolds with dimension $d > 1$, we will tackle it in Section 4.

## 4 Numerical Approach

In Section 3, we demonstrate that our proposed Riemannian Gaussian distribution can be used to achieve GDP in Theorem 3.1 and 3.2. Furthermore, we document our method in Algorithm 1 to compute the privacy budget $\mu$ in $S^1$. This and already existing Euclidean results encompass all the one-dimensional Riemannian manifolds. However, for general Riemannian manifolds of dimension

---

**Algorithm 1** Computing $\mu$ on $S^1$

---

**Input:** Sensitivity $\Delta \in (0, \pi]$, rate $\sigma$, number of $\varepsilon$ used $n_\varepsilon$
**Output:** $\mu$;

 1: Set $\varepsilon_{\max} = \pi \Delta / (2\sigma^2)$.
 2: **for** each $\varepsilon$ in $\{k, 2k, \dots, n_\varepsilon k\}$ where $k = \varepsilon_{\max} / n_\varepsilon$ **do**
 3:     Compute $l_\varepsilon = h(\sigma, \varepsilon, \Delta)$ as in Corollary 3.2.1 and $\mu_\varepsilon$ through $l_\varepsilon = \delta_\mu(\varepsilon)$.
 4: **end for**
 5: Compute $\mu = \max_\varepsilon \mu_\varepsilon$.
 6: **Return:** $\mu$.

---

$d > 1$, the integrals in inequality (2) are difficult to compute (see Section 3 for more explanation). One of the central difficulties is the dependence of the normalizing constant $Z(\eta, \sigma)$ on the footprint $\eta$ makes the expression of $A$ intractable. To avoid this dependence, we introduce the concept of homogeneous Riemannian manifolds.

**Definition 4.1** (Definition 4.6.1 in Berestovskii and Nikonorov [2020]). *A Riemannian manifold* $(\mathcal{M}, g)$ *is called a **homogeneous Riemannian manifold** if a Lie group $G$ acts transitively and isometrically on $\mathcal{M}$.*

Homogeneous Riemannian manifolds encompass a broad class of manifolds that are commonly encountered in statistics such as the (hyper)sphere [Bhattacharya and Bhattacharya, 2012, Mardia et al., 2000], the space of SPD Matrices[Pennec et al., 2019, Said et al., 2017, Hajri et al., 2016], the Stiefel manifold [Chakraborty and Vemuri, 2019, Turaga et al., 2008] and the Grassmann manifold [Turaga et al., 2008]. It's a more general class than Riemannian symmetric space, which is a common setting used in geometric statistics [Cornea et al., 2017, Asta, 2014, Said et al., 2018, Said, 2021, Chevallier et al., 2022]. Please refer to Appendix A.4 for more detail.

Informally, a homogeneous Riemannian manifold looks geometrically the same at every point. The transitive property required in the definition implies that any homogeneous Riemannian manifold $\mathcal{M}$ only has one orbit. Therefore, we have the following proposition.

**Proposition 4.1.** *If $\mathcal{M}$ is a homogeneous Riemannian manifolds, then $Z(\eta_1, y) = Z(\eta_2, y)$ for any $\eta_1, \eta_2 \in \mathcal{M}$.*

Therefore, on homogeneous Riemannian manifolds, we can simplify the set $A$ in Theorem 3.2 to $A = \left\{ y \in \mathcal{M} : d(\eta_2, y)^2 - d(\eta_1, y)^2 \geq 2\sigma^2 \varepsilon \right\}$. To further simplify (2), we will need a much stronger assumption than homogeneous Riemannian manifolds. In particular, we require the condition of constant curvature.

**Theorem 4.1** (Riemannian Gaussian Mechanism on Manifolds with Constant Curvature). *Let $\mathcal{M}$ be a Riemannian manifold with constant curvature and $f$ be a $\mathcal{M}$-valued summary with global sensitivity $\Delta$. The Riemannian Gaussian distribution with footpoint $f(\mathcal{D})$ and rate $\sigma > 0$ is $\mu$-GDP if and only if $\mu$ satisfies the following condition: $\exists \eta_1, \eta_2$ such that $d(\eta_1, \eta_2) = \Delta$ and $\forall \varepsilon \geq 0$*

$$\int_A p_{\eta_1, \sigma}(y) \, d\nu(y) - e^\varepsilon \int_A p_{\eta_2, \sigma}(y) \, d\nu(y) \leq \delta_\mu(\varepsilon) \tag{3}$$

*where $A = \left\{ y \in \mathcal{M} : d(\eta_2, y)^2 - d(\eta_1, y)^2 \geq 2\sigma^2 \varepsilon \right\}$.*

*Proof.* See Appendix A.5. $\qquad \square$

Theorem 4.1 tells us that instead of evaluating the integrals of 2 on every neighboring pairs $\eta_1$ and $\eta_2$, we only need to check one such pair. Despite the convenience it provides us, evaluating the integrals remains a challenge, even for an elementary space like $S^d$ with $d > 1$. To circumvent this challenge, we employ the MCMC technique for the integral computations. The main idea is simple: Let $Y_1, Y_2, \dots, Y_n$ be independent and identically distributed random variables from $N_{\mathcal{M}}(\eta, \sigma^2)$, and denotes $Z_i = \mathbf{1} \left\{ \frac{1}{2\sigma^2} \left( -d(\eta_1, Y_i)^2 + d(\eta_2, Y_i)^2 \right) \geq \varepsilon \right\}$. By the strong law of large number [Dudley, 2002], we then have

$$\frac{1}{n} \sum_{i=1}^n Z_i \to \int_A p_{\eta, \sigma}(y) d\nu(y) \quad \text{a.s..}$$

By using $\frac{1}{n}\sum_{i=1}^{n} Z_i$ as an approximation, we avoid the challenge of evaluating the integrals analytically. The detailed algorithm is documented in Algorithm 2. It's known that any space of constant curvature is isomorphic to one of the spaces: Euclidean space, sphere, and hyperbolic space [Vinberg et al., 1993, Woods, 1901]. Therefore, Algorithm 2 offers a straightforward and practical method for assessing the privacy budget $\mu$ on spheres and hyperbolic spaces. Furthermore, Algorithm 2 can be extended to a more general class of Riemannian manifold by sampling more than one pair of $\eta, \eta'$ in step 1. The determination of the number of pairs being sampled and the selection method to ensure sufficient dissimilarity among the sampled pairs is an aspect that requires further investigation.

---

**Algorithm 2** Computing $\mu$ on Manifolds with Constant Curvature

---

**Input:** Sensitivity $\Delta$, rate $\sigma^2$, Monte Carlo sample size $n$, number of $\varepsilon$ used $n_\varepsilon$, maximum $\varepsilon$ used $\varepsilon_{\max}$, number of MCMC samples $m$
**Output:** privacy budget $\mu$;
  1: Sample a random point $\eta$ on $\mathcal{M}$ and another point $\eta'$ on the sphere center at $\eta$ with radius $\Delta$.
  2: **for** each $j$ in $1, 2, \ldots, m$ **do**
  3:     Sample $y_{1j}, \ldots, y_{nj} \sim N_{\mathcal{M}}(\eta, \sigma^2), y'_{1j}, \ldots, y'_{nj} \sim N_{\mathcal{M}}(\eta', \sigma^2)$.
  4:     Compute $d_{ij} = d(\eta', y_{ij})^2 - d(\eta, y_{ij})^2$ and $d'_{ij} = d(\eta', y'_{ij})^2 - d(\eta, y'_{ij})^2$ for $i$ in $1, 2, \ldots, n$.
  5:     **for** each $\varepsilon$ in $\{k, 2k, \ldots, n_\varepsilon k\}$ where $k = \varepsilon_{\max}/n_\varepsilon$ **do**
  6:        Compute $l_\varepsilon^{(j)} = \sum_{i=1}^{n} \mathbf{1}(d_{ij} \geq 2\sigma^2\varepsilon)/n - e^\varepsilon \sum_{i=1}^{n} \mathbf{1}(d'_{ij} \geq 2\sigma^2\varepsilon)/n$.
  7:     **end for**
  8: **end for**
  9: Compute $l_\varepsilon = \sum_{j=1}^{m} l_\varepsilon^{(j)}/m$ and $\mu_\varepsilon$ via $l_\varepsilon = \delta_\mu(\varepsilon)$ for each $\varepsilon$. Compute $\mu = \max_\varepsilon \mu_\varepsilon$.
10: **Return**: $\mu$.

---

### 4.1 Sampling from Riemannian Gaussian Distribution

The one crucial step in Algorithm 2 involves sampling from the two Riemannian Gaussian distributions $N_{\mathcal{M}}(\eta, \sigma^2)$ and $N_{\mathcal{M}}(\eta', \sigma^2)$. Since their densities (1) are known up to a constant, a Metropolis-Hasting algorithm would be a natural choice. In this section, we describe a general Metropolis-Hasting algorithm for sampling from a Riemannian Gaussian distribution on an arbitrary homogeneous Riemannian manifold [Pennec et al., 2019]. However, there are more efficient sampling algorithms that are tailored to specific manifolds (e.g., [Said et al., 2017, Hauberg, 2018]).

The Metropolis-Hasting algorithm involves sampling a candidate $y$ from a proposal distribution $q(\cdot|x)$. The acceptance probability of accepting $y$ as the new state is $\alpha(x, y) = \min\{1, q(y|x)p_{\eta,\sigma}(y)/[q(x|y)p_{\eta,\sigma}(x)]\}$. A natural choice for the proposal distribution $q(\cdot|x)$ could be an exponential-wrapped Gaussian distribution [Galaz-Garcia et al., 2022, Chevallier et al., 2022]. Informally, it's the distribution resulting from "wrapping" a Gaussian distribution on tangent space back to the manifold using the exponential map. Given the current state $x \in \mathcal{M}$, we sample a tangent vector $v \sim N(\mathbf{0}, \sigma^2\mathbf{I})$ on the tangent space $T_x\mathcal{M}$. If $\|v\|$ is less than the injectivity radius, we then accept the newly proposed state $y = \exp_x(v)$ with probability $\alpha(x, y)$. Please refer to Section 2.5 in Pennec et al. [2019] for the detailed algorithm.

### 4.2 GDP on $\mathbb{R}$ and $S^1$

To evaluate the performance of Algorithm 2, we will conduct simulations on Euclidean space $\mathbb{R}$ and unit circle $S^1$ as the relation between $\mu$ and $\sigma$ is established in Dong et al. [2022] and Corollary 3.2.1.

We fix sensitivity $\Delta = 1$ and let $\sigma = k/4$ with $1 \leq k \leq 16$. For each $\sigma$, we determine the privacy budget $\mu$ using two approaches: (i) using Algorithm 1 with $n_\varepsilon$ set as 1000 for $S^1$ and using $\mu = 1/\sigma$ for $\mathbb{R}$; (ii) using Algorithm 2 with $n = 1000, n_\varepsilon = 1000, m = 100, \varepsilon_{\max} = \pi/(2\sigma^2)$ for $S^1$ and $\varepsilon_{\max} = \max\{10, 5/\sigma + 1/(2\sigma^2)\}$ for $\mathbb{R}$. Since Algorithm 2 is a randomized algorithm, we generate 20 replicates for approach (ii) for each $\sigma$. In Figure 1, the first panel plots the sample means of the $\mu$ generated by approach (ii) (in grey with rectangular symbols) with error bars indicating the minimum & the maximum of the $\mu$'s and the $\mu$ computed by approach (i) (in red with circular symbols). Additionally, as a comparison, we also plot the $\mu = 1/\sigma$ for Euclidean space (in blue with triangular symbols) in the first plot. As we see from the first plot, the GDP results on $S^1$ are almost exactly the same as on $\mathbb{R}$ for smaller $\sigma$. This is expected as manifolds are locally Euclidean,

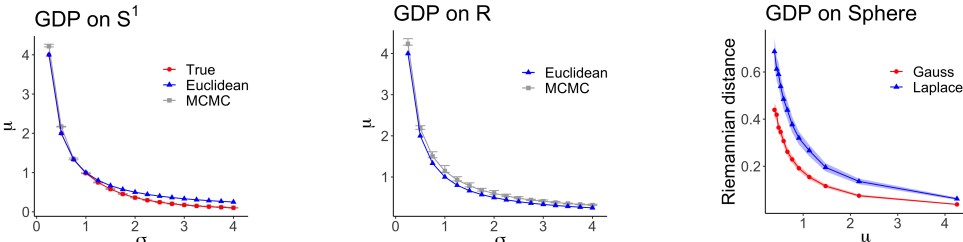

Figure 1: **First and Second plots**: Red lines with circular symbols represent the relation between privacy budget $\mu$ and rate $\sigma$ on the unit circle $S^1$. Blue lines with triangular symbols represent the relation in Euclidean space. Gray lines with rectangular symbols plot the sample mean of the $\mu$, across the 20 repeats, computed at a variety of $\sigma$ using Algorithm 2. The error bar indicates the minimum and maximum of the $\mu$'s. Refer to Section 4.2 for details. **Third plot**: Blue line with triangular symbols indicates the sample mean, across 100 repeats, of the Riemannian distances $d(\bar{x}, \bar{x}_{\text{laplace}})$, while the red line with circular symbols indicates the sample mean of the Riemannian distances $d(\bar{x}, \bar{x}_{\text{gauss}})$. The error bands indicate the sample mean $\pm 4$SE. Refer to Section 5.2 for details.

and the smaller the $\sigma$ the closer the results will be. As the $\sigma$ gets larger, the privacy budget $\mu$ gets smaller on $S^1$ compared to on $\mathbb{R}$. This can be explained by the fact that $S^1$ is a compact space while $\mathbb{R}$ is not. For the second panel, we plot the exactly same things for Euclidean spaces. As we can observe from both panels, our Algorithm 2 gives a fairly accurate estimation for larger $\sigma$. However, as $\sigma$ gets smaller, Algorithm 2 has a tendency to generate estimates that exhibit a higher degree of overestimation.

# 5 Simulations

In this section, we evaluate the utility of our Riemannian Gaussian mechanism by focusing on the task of releasing differentially private Fréchet mean. Specifically, we conduct numerical examples on the unit sphere, which is commonly encountered in statistics [Bhattacharya and Bhattacharya, 2012, Mardia et al., 2000]. As a comparison to our Riemannian Gaussian mechanism, we use the Riemannian Laplace mechanism implemented in Reimherr et al. [2021] to achieve GDP. Although the Riemannian Laplace mechanism is developed originally to achieve $\varepsilon$-DP, it's shown in Liu et al. [2022] any mechanism that satisfies $\varepsilon$-DP can achieve $\mu$-GDP with $\mu = -2\Phi^{-1}\left(1/(1 + e^{\varepsilon})\right) \leq \sqrt{\pi/2}\varepsilon$. Our results show significant improvements in utility for the privatization of the Fréchet mean when using our proposed Gaussian mechanism, as opposed to the Laplace mechanism, in both examples. In Section 5.1, we cover some basics on differentially private Fréchet mean. In Section 5.2, we discuss the numerical results on the sphere. Simulations are done in R on a Mac Mini computer with an Apple M1 processor with 8 GB of RAM running MacOS 13. For more details on each simulation, please refer to Appendix A.6. The R code is available in the GitHub repository: `https://github.com/Lei-Ding07/Gaussian-Differential-Privacy-on-Riemannian-Manifolds`

## 5.1 Differentially Private Fréchet Mean

For more details on Fréchet mean under the DP setting, please refer to Reimherr et al. [2021]. Consider a set of data $x_1, \ldots, x_N$ on $\mathcal{M}$. The Euclidean sample mean can be generalized to Riemannian manifolds as the sample Fréchet mean, which is the minimizer of the sum-of-squared distances to the data, $\bar{x} = \arg\min_{x \in \mathcal{M}} \sum_{i=1}^{N} d(x, x_i)^2$. To ensure the existence & uniqueness of the Fréchet mean and to determine its sensitivity, we need the following assumption.

**Assumption 1.** *The data $\mathcal{D} \subseteq B_r(m_0)$ for some $m_0 \in \mathcal{M}$, where $r < r^*$ with $r^* = \min\{\text{inj}\,\mathcal{M}, \pi/(2\sqrt{\kappa})\}/2$ for $\kappa > 0$ and $r^* = \text{inj}\,\mathcal{M}/2$ for $\kappa \leq 0$. Note $\kappa$ denotes an upper bound on the sectional curvatures of $\mathcal{M}$.*

Under Assumption 1, we can then compute the sensitivity of the Fréchet mean [Reimherr et al., 2021]. consider two datasets $\mathcal{D} \simeq \mathcal{D}'$. If $\bar{x}$ and $\bar{x}'$ are the two sample Fréchet means of $\mathcal{D}$ and $\mathcal{D}'$

respectively, then

$$d\left(\bar{x}, \bar{x}'\right) \leq \frac{2r(2 - h(r,\kappa))}{nh(r,\kappa)}, \quad h(r,\kappa) = \begin{cases} 2r\sqrt{\kappa}\cot(\sqrt{\kappa}2r) & \kappa > 0 \\ 1 & \kappa \leq 0 \end{cases}$$

## 5.2 Sphere

First, we revisit some background materials on spheres, refer to Bhattacharya and Bhattacharya [2012], Reimherr et al. [2021] for more details. We denote the $d$-dimensional unit sphere as $S^d$ and identify it as a subspace of $\mathbb{R}^{d+1}$ as $S^d = \{p \in \mathbb{R}^{d+1} : \|p\|_2 = 1\}$. Similarly, at each $p \in S^d$, we identify the tangent space $T_p S^d$ as $T_p S^d = \{v \in \mathbb{R}^{d+1} : v^\top p = 0\}$. The geodesics are the great circles, $\gamma_{p,v}(t) = \cos(t)p + \sin(t)v$ with $-\pi < t \leq \pi$ where $\gamma_{p,v}$ denotes the geodesic starts at $p$ with unit direction vector $v$. The exponential map $\exp_p : T_p S^d \to S^d$ is given by $\exp_p(0) = p$ and $\exp_p(v)) := \cos(\|v\|)p + \sin(v)v/\|v\|$ for $v \neq 0$. The inverse of the exponential map $\log_p : S^d \setminus \{-p\} \to T_p S^d$ has the expression $\log_p(p) = 0$ and $\log_p(q) = \arccos(p^\top q)[q - (p^\top q)p]\left[1 - (p^\top q)^2\right]^{-1/2}$ for $q \neq p, -q$. It follows that the distance function is given by $d(p,q) = \arccos(p^\top q) \in [0,\pi]$. Therefore, $S^d$ has an injectivity radius of $\pi$.

We initiate our analysis by generating sample data $\mathcal{D} = \{x_1, \ldots, x_n\}$ from a ball of radius $\pi/8$ on $S^2$ and subsequently computing the Fréchet mean $\bar{x}$. To disseminate the private Fréchet mean, we implement two methods: (i) We first generate the privatized mean $\bar{x}_{\text{gauss}}$ by drawing from $N_{\mathcal{M}}(\bar{x}, \sigma^2)$ employing the sampling method proposed by Hauberg [2018]. The privacy budget $\mu$ is then computed using Algorithm 2. (ii) Next, we convert $\mu$-GDP to the equivalent $\varepsilon$-DP using $\varepsilon = \log[1 - \Phi(-u/2))/\Phi(-u/2)]$, and generate the privatized mean $\bar{x}_{\text{laplace}}$ by sampling from the Riemannian Laplace distribution with footprint $\bar{x}$ and rate $\Delta/\varepsilon$ using the sampling method introduced by You and Shung [2022].

Throughout these simulations, we fix the sample size at $n = 10$ to maintain a constant sensitivity $\Delta$. With $\Delta$ held constant, we let the rate $\sigma = k/4$ with $1 \leq k \leq 12$. The objective here is to discern the difference between the two distances $d(\bar{x}, \bar{x}_{\text{gauss}})$ and $d(\bar{x}, \bar{x}_{\text{laplace}})$ across varying privacy budgets $\mu$. The third plot in Figure 1 displays the sample mean of the Riemannian distances $d(\bar{x}, \bar{x}_{\text{gauss}})$ (in red with circular symbols) and $d(\bar{x}, \bar{x}_{\text{laplace}})$ (in blue with triangular symbols) across 1000 iterations with the error band indicating the sample mean $\pm 4$SE. From observing the third plot, we see that our Gaussian mechanism achieves better utility, especially with a smaller privacy budget $\mu$. With larger $\mu$, the gain in utility is less pronounced. One obvious reason is that there are much fewer perturbations with larger $\mu$ for both approaches, so the difference is subtle. The other reason is that Algorithm 2 has a tendency to overestimate $\mu$ with smaller $\sigma$. Effectively, $\bar{x}_{\text{gauss}}$ satisfies $\mu$-GDP with a smaller $\mu$ compared to $\bar{x}_{\text{laplace}}$.

## 6 Conclusions and Future Directions

In this paper, we extend the notion of GDP over general Riemannian manifolds. Then we showed that GDP can be achieved when using Riemannian Gaussian distribution as the additive noises. Furthermore, we propose a general MCMC-based algorithm to compute the privacy budget $\mu$ on manifolds with constant curvature. Lastly, we show through simulations that our Gaussian mechanism outperforms the Laplace mechanism in achieving $\mu$-GDP on the unit sphere $S^d$.

There are many future research directions. First of all, the framework established in this paper can be used for extending $(\varepsilon, \delta)$-DP to general Riemannian manifolds. There are several points of improvement around Algorithm 2 as well. Although Algorithm 2 provides us a general method of computing the privacy budget $\mu$, it lacks an error bound on its estimation. Furthermore, due to the random nature of Algorithm 2, a variance estimator of the output is desirable and can better inform the end user. Though we demonstrate the utility of our mechanism through simulation in Section 5, it's difficult to obtain a theoretical utility guarantee due to the lack of simple analytical relation between $\mu$ and rate $\sigma$. Furthermore, although Algorithm 1 requires the manifolds to have constant curvature, it's possible to extend Algorithm 1 and Theorem 4.1 to a slightly more general class of manifolds. Additionally, the Riemannian Gaussian distribution defined in this paper is not the only way of extending Gaussian distribution to Riemannian manifolds as mentioned in Section 3. Potentially, the two other approaches, the wrapped distribution approach, and the heat kernel approach can also be

used to achieve GDP on Riemannian manifolds as well. In particular, there are many rich results around heat kernel on Riemannian manifolds (see Grigoryan [2009] for example). Incorporating the heat kernel in a privacy mechanism presents ample potential for novel and noteworthy discoveries.

## Acknowledgements

We would like to express our greatest appreciation to Professor Xiaowei Wang of Rutgers University and Professor Xi Chen from the University of Alberta for their invaluable guidance, support, discussion, and expertise throughout the course of this research. This work was supported by the Canada CIFAR AI Chairs program, the Alberta Machine Intelligence Institute, the Natural Sciences and Engineering Council of Canada (NSERC), the Canada Research Chair program from NSERC, and the Canadian Statistical Sciences Institute. We also thank Qirui Hu from Tsinghua University and all the constructive suggestions and comments from the reviewers.

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

# A  Appendix

## A.1  Theorem 3.1

First, we need to introduce the notion of privacy profile [Balle et al., 2018]:

The **privacy profile** $\delta_{\mathcal{M}}$ of a mechanism $\mathcal{M}$ is a function associating to each privacy parameter $\alpha = e^{\varepsilon}$ a bound on the $\alpha$-divergence between the results of running the mechanism on two adjacent datasets, i.e. $\delta_{\mathcal{M}}(\varepsilon) = \sup_{x \simeq x'} D_{e^{\varepsilon}}(\mathcal{M}(x)\|\mathcal{M}(x'))$ where the $\alpha$-divergence ($\alpha \geq 1$) between two probability measures $\mu, \mu'$ is defined as

$$D_{\alpha}(\mu\|\mu') = \sup_{E}(\mu(E) - \alpha\mu'(E)) = \int_{Z}\left[\frac{d\mu}{d\mu'}(z) - \alpha\right]_{+} d\mu'(z) = \sum_{z \in Z}[\mu(z) - \alpha\mu'(z)]_{+},$$

where $E$ ranges over all measurable subsets of $Z, [\cdot]_{+} = \max\{\cdot, 0\}$. [2] Informally speaking, the privacy profile represents the set of all of the privacy parameters under which a mechanism provides differential privacy. Furthermore, the privacy profile can be computed using Theorem 5 of Balle and Wang [2018]:

$$\delta_{\mathcal{M}}(\varepsilon) = \sup_{\mathcal{D} \simeq \mathcal{D}'}\left(\Pr\left[L_{\mathcal{M}}^{\mathcal{D},\mathcal{D}'} > \varepsilon\right] - e^{\varepsilon}\Pr\left[L_{\mathcal{M}}^{\mathcal{D}',\mathcal{D}} < -\varepsilon\right]\right)$$

where $L_{\mathcal{M}}^{\mathcal{D},\mathcal{D}'}$ is the **privacy loss** random variable of the mechanism $\mathcal{M}$ on inputs $\mathcal{D} \simeq \mathcal{D}'$ defined as $L_{\mathcal{M}}^{\mathcal{D},\mathcal{D}'} = \log(d\mu/d\mu')(\mathbf{z})$, where $\mu = \mathcal{M}(\mathcal{D}), \mu' = \mathcal{M}(\mathcal{D}')$, and $\mathbf{z} \sim \mu$. It follows that for our Riemannian Gaussian mechanism $\mathcal{M}$, the privacy profile $\delta_{\mathcal{M}}$ can be rewritten as

$$\delta_{\mathcal{M}} = \sup_{\mathcal{D} \simeq \mathcal{D}'}\int_{A} p_{\eta_1,\sigma}(y)\, d\nu(y) - e^{\varepsilon}\int_{A} p_{\eta_2,\sigma}(y)\, d\nu(y)$$

where $A := \{y \in \mathcal{M} : p_{\eta_1,\sigma}(y)/p_{\eta_2,\sigma}(y) \geq e^{\varepsilon}\}$ and $\eta_1 := f(\mathcal{D}), \eta_2 := f(\mathcal{D}')$. This is exactly the left-hand side of (2).

The following theorem establishes a connection between GDP and privacy profile:

**Theorem A.1** (Theorem 3.3 of Liu et al. [2022]). *Let* $\mu_0 := \sqrt{\lim_{\varepsilon \to +\infty}\frac{\varepsilon^2}{-2\log \delta_{\mathcal{A}}(\varepsilon)}}$. *A privacy mechanism $\mathcal{A}$ with the privacy profile $\delta_{\mathcal{A}}(\varepsilon)$ is $\mu$-GDP if and only if $\mu_0 < \infty$ and $\mu$ is no smaller than $\mu_0$.*

Theorem A.1 implies that a mechanism with finite $\mu_0$ is $\mu$-GDP for some privacy budget $\mu$. Note that Theorem 3.1 only tells us that Riemannian Gaussian distribution can be used to achieve $\mu$-GDP for some $\mu$. Therefore, to prove it, we only need to show that $\mu_0$ is finite. We will show $\mu_0 < \infty$ by demonstrate that $-\log \delta_{\mathcal{A}}(\varepsilon) = O(\varepsilon^2)$. To do so, we will need the Bishop-Gromov comparison theorem:

**Theorem A.2** (Bishop-Gromov; Lemma 7.1.4 in Petersen [2006]). *Let $M$ be a complete $n$-dimensional Riemannian manifold whose Ricci curvature satisfies the lower bound*

$$\mathrm{Ric} \geq (n-1)K$$

*for a constant $K \in \mathbb{R}$. Let $M_K^n$ be the complete $n$-dimensional simply connected space of constant sectional curvature $K$ (and hence of constant Ricci curvature $(n-1)K$). Denote by $B(p,r)$ the ball of radius $r$ around a point $p$, defined with respect to the Riemannian distance function. Then, for any $p \in M$ and $p_K \in M_K^n$, the function*

$$\phi(r) = \frac{\mathrm{Vol}\, B(p,r)}{\mathrm{Vol}\, B(p_K,r)}$$

*is non-increasing on $(0,\infty)$. As $r$ goes to zero, the ratio approaches one, so together with the monotonicity this implies that*

$$\mathrm{Vol}\, B(p,r) \leq \mathrm{Vol}\, B(p_K,r).$$

Bishop-Gromov comparison theorem not only gives us the control of volume growth of certain manifolds but also gives a rough classification by sectional curvature. Besides, this is a global property in the sense that $p$ and $p_K$ can be arbitrary points on the manifolds.

---

[2]It is known that a mechanism $\mathcal{M}$ is $(\varepsilon, \delta)$-DP if and only if $D_{e^{\varepsilon}}(\mathcal{M}(\mathcal{D})\|\mathcal{M}(\mathcal{D}')) \leq \delta$ for every $\mathcal{D}$ and $\mathcal{D}'$ such that $\mathcal{D} \simeq \mathcal{D}'$.

### A.1.1 Proof of Theorem 3.1

*Proof.* By A.1, we only need to show that for any $\eta \in \mathcal{M}$, when $\varepsilon \to \infty$,

$$\int_A p_{\eta,\sigma}(y)d\nu(y) = e^{-O(\varepsilon^2)}$$

where $A$ is given by

$$A = \{y \in \mathcal{M}|p_{\eta,\sigma}(y)/p_{\eta',\sigma}(y) > e^\varepsilon\}$$

Let's consider $\mathcal{M}\backslash A = \{y \in \mathcal{M} : p_{\eta,\sigma}(y)/p_{\eta',\sigma}(y) \le e^\varepsilon\}$. We have

$$\log\left(\frac{p_{\eta,\sigma}(y)}{p_{\eta',\sigma}(y)}\right)$$

$$= \frac{1}{2\sigma^2}(d(\eta,y)^2 - d(\eta',y)^2) + C$$

$$\le \frac{\Delta}{2\sigma^2}(2d(\eta,y) + \Delta) + C, \quad \text{by triangular inequality,}$$

where $C = \log(Z(\eta,\sigma)) - \log(Z(\eta',\sigma))$. Thus we have,

$$d(\eta,y) \le \frac{2\sigma^2(\varepsilon - C) - \Delta^2}{2\Delta} \implies \frac{p_{\eta,\sigma}(y)}{p_{\eta',\sigma}(y)} \le e^\varepsilon$$

Let $r = \frac{2\sigma^2(\varepsilon-C)-\Delta^2}{2\Delta}$, note that since $B_\eta(r) \subseteq \mathcal{M}\backslash A$, we have $A \subseteq \mathcal{M}\backslash B_\eta(r)$. Thus, we only need to prove the following:

$$\int_{\mathcal{M}\backslash B_\eta(r)} p_{\eta,\sigma^2}(y)d\nu(y) = e^{-O(\varepsilon^2)}$$

when $\varepsilon \to \infty$. One can easily show the following inequality,

$$\int_{B_\eta(2r)\backslash B_\eta(r)} p_{\eta,\sigma}(y)d\nu(y) \le Z(\eta,\sigma)\, e^{-\frac{r^2}{\sigma^2}}(\operatorname{Vol} B(\eta,2r) - \operatorname{Vol} B(\eta,r)) \tag{4}$$

By Theorem A.2, we have the following three cases:

1. $K > 0$. Then the standard space $M_K^n$ is the $n$-sphere of radius $1/\sqrt{K}$. $(\operatorname{Vol} B(\eta,2r) - \operatorname{Vol} B(\eta,r))$ is obviously less than the volume of the whole space $M_K^n$. Thus we have

$$\int_{B_\eta(2r)\backslash B_\eta(r)} p_{\eta,\sigma}(y)d\nu(y) \le Z(\eta,\sigma)\, e^{-\frac{r^2}{\sigma^2}} s(n)\sqrt{K}^{1-n} \tag{5}$$

   where $s(n) = \frac{2\pi^{\frac{n}{2}}}{\Gamma(\frac{n}{2})}$ is a constant relative to the dimension $n$. One can easily find that as $\varepsilon \to \infty$, $B_\eta(2r)$ will cover the $M_K^n$, and the right-hand side of inequality 5 will approach to 0 as $e^{-O(\varepsilon^2)}$.

2. $K = 0$. Then the standard space $M_K^n$ is the $n$ dimensional Euclidean space $\mathbb{R}^n$. We have

$$\int_{B_\eta(2r)\backslash B_\eta(r)} p_{\eta,\sigma}(y)d\nu(y) \le Z(\eta,\sigma)\, e^{-\frac{r^2}{\sigma^2}} \frac{\pi^{n/2}r^n}{\Gamma(\frac{n}{2}+1)}. \tag{6}$$

   The same with above, when $\varepsilon \to \infty$, $B_\eta(2r)$ will cover the $\mathbb{R}^n$ and the right-hand side of 6 will approach to 0 as $e^{-O(\varepsilon^2)}$.

3. $K < 0$. The standard space $M_n(K)$ is the hyperbolic $n$-space $\mathbb{H}^n$. The hyperbolic volume $\operatorname{Vol} B(\eta_K, r)$ with any $\eta_k \in \mathbb{H}^n$ is given by

$$\operatorname{Vol} B(\eta_K, r) = s(n)\int_0^r \left(\frac{\sinh(\sqrt{-K}t)}{\sqrt{-K}}\right)^{n-1} dt \tag{7}$$

   where the hyperbolic function is given by $\sinh(x) = (e^x - e^{-x})/2$. It's not hard to see that

$$\sinh^n(t) \le \frac{(n+1)e^{nt}}{2^n}. \tag{8}$$

Plugging the 8 into 7, we have

$$\text{Vol } B(\eta_K, r) \le s_n \frac{n}{(n-1)2^{n-1}\sqrt{-K}^n} e^{\sqrt{-K}(n-1)r}. \tag{9}$$

Combining 9 and 4, we have

$$\int_{B_\eta(2r)\setminus B_\eta(r)} p_{\eta,\sigma}(y)d\nu(y) \le c(n) \, Z(\eta,\sigma) \, e^{-\frac{r^2}{\sigma^2}+\sqrt{-K}(n-1)2r}. \tag{10}$$

The principle part of the exponent of the right-hand side of 10 is still $-\varepsilon^2$. Thus, when $\varepsilon \to \infty$, it approaches to 0 as $e^{-O(\varepsilon^2)}$.

$\square$

## A.2 Theorem 3.2

*Proof.* Follows directly from Definition 2.2, Theorem 3.1 and Theorem 5 in Balle and Wang [2018].
$\square$

## A.3 Corollary 3.2.1

*Proof.* In this proof, we will parameterize points on $S^1$ using their polar angles.

On $S^1$, the Riemannian Gaussian distribution with footprint $\eta$ and rate $\sigma$ has the following density,

$$p_{\eta,\sigma}(\theta) = \frac{1}{Z_\sigma} e^{-\frac{1}{2\sigma^2}(\theta-\eta \mod \pi)^2}, \quad Z_\sigma = \sqrt{2\pi}\sigma \left[ \Phi\left(\frac{\pi}{\sigma}\right) - \Phi\left(-\frac{\pi}{\sigma}\right) \right].$$

Note that since $S^1$ has constant curvature, we can use Theorem 4.1 instead of Theorem 3.2

WLOG we assume $\eta_1 = 2\pi - \frac{\Delta}{2}$ and $\eta_2 = \frac{\Delta}{2}$ and thus $d(\eta_1, \eta_2) = \Delta$. Given an arbitrary $\varepsilon$, the set $A$ takes the following form,

$$A = \left[ \pi + \frac{\sigma^2\varepsilon}{\Delta}, 2\pi - \frac{\sigma^2\varepsilon}{\Delta} \right].$$

and it follows that we must have

$$\varepsilon \in [0, \pi\Delta/(2\sigma^2)]. \tag{11}$$

Thus we have

$$\int_A p_{\eta_1,\sigma}(y) \, d\nu(y) - e^\varepsilon \int_A p_{\eta_2,\sigma}(y) \, d\nu(y)$$

$$= \frac{1}{Z_\sigma} \left[ \int_A e^{-\frac{1}{2\sigma^2}(\eta_1-\theta \mod \pi)^2} d\theta - e^\varepsilon \int_A e^{-\frac{1}{2\sigma^2}(\eta_2-\theta \mod \pi)^2} d\theta \right]$$

$$= \frac{1}{Z_\sigma} \left[ \int_{\pi+\frac{\sigma^2\varepsilon}{\Delta}}^{2\pi-\frac{\sigma^2\varepsilon}{\Delta}} e^{-\frac{1}{2\sigma^2}(\eta_1-\theta \mod \pi)^2} d\theta - e^\varepsilon \int_A e^{-\frac{1}{2\sigma^2}(\eta_2-\theta \mod \pi)^2} d\theta \right]$$

$$= \frac{1}{Z_\sigma} \left[ \int_{\pi+\frac{\sigma^2\varepsilon}{\Delta}}^{2\pi-\frac{\sigma^2\varepsilon}{\Delta}} e^{-\frac{1}{2\sigma^2}(\eta_1-\theta)^2} d\theta - e^\varepsilon \int_A e^{-\frac{1}{2\sigma^2}(\eta_2-\theta \mod \pi)^2} d\theta \right]$$

$$= \frac{1}{Z_\sigma} \left[ \Phi\left( \frac{2\pi}{\sigma} - \frac{\sigma\varepsilon}{\Delta} - \frac{\eta_1}{\sigma} \right) - \Phi\left( \frac{\pi}{\sigma} + \frac{\sigma\varepsilon}{\Delta} - \frac{\eta_1}{\sigma} \right) - e^\varepsilon \int_A e^{-\frac{1}{2\sigma^2}(\eta_2-\theta \mod \pi)^2} d\theta \right]$$

$$= \frac{1}{Z_\sigma} \left[ \Phi\left( -\frac{\sigma\varepsilon}{\Delta} + \frac{\Delta}{2\sigma} \right) - \Phi\left( \frac{\sigma\varepsilon}{\Delta} + \frac{\Delta}{2\sigma} - \frac{\pi}{\sigma} \right) - e^\varepsilon \int_A e^{-\frac{1}{2\sigma^2}(\eta_2-\theta \mod \pi)^2} d\theta \right].$$

We have evaluated the first integral, and now let's consider the second integral. For $\varepsilon \leq \frac{\Delta^2}{2\sigma^2}$, we have

$$\int_A e^{-\frac{1}{2\sigma^2}(\mu_2 - \theta \mod \pi)^2} d\theta$$

$$= \int_{\pi + \frac{\sigma^2\varepsilon}{\Delta}}^{\mu_2 + \pi} e^{-\frac{1}{2\sigma^2}(\theta - \mu_2)^2} d\theta + \int_{\mu_2 + \pi}^{2\pi - \frac{\sigma^2\varepsilon}{\Delta}} e^{-\frac{1}{2\sigma^2}(\theta - (2\pi + \mu_2))^2} d\theta$$

$$= \sqrt{2\pi}\sigma \left[ \Phi\left(\frac{\pi}{\sigma}\right) - \Phi\left(\frac{\pi}{\sigma} + \frac{\sigma\varepsilon}{\Delta} - \frac{\Delta}{2\sigma}\right) + \Phi\left(-\frac{\sigma\varepsilon}{\Delta} - \frac{\Delta}{2\sigma}\right) - \Phi\left(-\frac{\pi}{\sigma}\right) \right].$$

Thus for $\varepsilon \leq \frac{\Delta^2}{2\sigma^2}$ we have,

$$\int_A p_{\eta_1,\sigma}(y) \, d\nu(y) - e^{\varepsilon} \int_A p_{\eta_2,\sigma}(y) \, d\nu(y)$$

$$= \frac{\sqrt{2\pi}\sigma}{Z_\sigma} \left[ \Phi\left(-\frac{\sigma\varepsilon}{\Delta} + \frac{\Delta}{2\sigma}\right) - e^{\varepsilon}\Phi\left(-\frac{\sigma\varepsilon}{\Delta} - \frac{\Delta}{2\sigma}\right) \right] \tag{12}$$

$$- \frac{\sqrt{2\pi}\sigma}{Z_\sigma} \left[ \Phi\left(\frac{\sigma\varepsilon}{\Delta} + \frac{\Delta}{2\sigma} - \frac{\pi}{\sigma}\right) - e^{\varepsilon}\Phi\left(\frac{\sigma\varepsilon}{\Delta} - \frac{\Delta}{2\sigma} + \frac{\pi}{\sigma}\right) \right]$$

$$- e^{\varepsilon}.$$

Similarly, for $\varepsilon > \frac{\Delta^2}{2\sigma^2}$, we have,

$$\int_A e^{-\frac{1}{2\sigma^2}(\mu_2 - \theta \mod \pi)^2} d\theta$$

$$= \int_{\pi + \frac{\sigma^2\varepsilon}{\Delta}}^{2\pi - \frac{\sigma^2\varepsilon}{\Delta}} e^{-\frac{1}{2\sigma^2}(\theta - (2\pi + \mu_2))^2} d\theta$$

$$= \sqrt{2\pi}\sigma \left[ \Phi\left(-\frac{\sigma\varepsilon}{\Delta} - \frac{\Delta}{2\sigma}\right) - \Phi\left(-\frac{\pi}{\sigma} + \frac{\sigma\varepsilon}{\Delta} - \frac{\Delta}{2\sigma}\right) \right].$$

Thus for $\varepsilon > \frac{\Delta^2}{2\sigma^2}$ we have,

$$\int_A p_{\eta_1,\sigma}(y) \, d\nu(y) - e^{\varepsilon} \int_A p_{\eta_2,\sigma}(y) \, d\nu(y)$$

$$= \frac{\sqrt{2\pi}\sigma}{Z_\sigma} \left[ \Phi\left(-\frac{\sigma\varepsilon}{\Delta} + \frac{\Delta}{2\sigma}\right) - e^{\varepsilon}\Phi\left(-\frac{\sigma\varepsilon}{\Delta} - \frac{\Delta}{2\sigma}\right) \right] \tag{13}$$

$$- \frac{\sqrt{2\pi}\sigma}{Z_\sigma} \left[ \Phi\left(\frac{\sigma\varepsilon}{\Delta} + \frac{\Delta}{2\sigma} - \frac{\pi}{\sigma}\right) - e^{\varepsilon}\Phi\left(\frac{\sigma\varepsilon}{\Delta} - \frac{\Delta}{2\sigma} - \frac{\pi}{\sigma}\right) \right].$$

Put (11), (12) and (13) together with Theorem 4.1, we have proved Corollary 3.2.1. $\qquad \square$

## A.4   Homogeneous Riemannian Manifolds

For more detailed treatment on homogenous Riemannian manifolds and related concepts, refers to Helgason [1962], Berestovskii and Nikonorov [2020], Lee [2006] for details and Chakraborty and Vemuri [2019] for a more concise summary.

### A.4.1   Group actions on Manifolds

In this section, we will introduce some basic facts about group action which will be used to introduce homogeneous Riemannian manifolds in later section. The materials covered in this section can be found in any standard Abstract Algebra texts.

**Definition A.1.** *A **group** $(G, \cdot)$ is a non-empty set $G$ together with a binary operation $\cdot : G \times G \to G, (a, b) \mapsto a \cdot b$ such that the following three axioms are satisfied:*

- ***Associativity:*** *$\forall a, b, c \in G, (a \cdot b) \cdot c = a(b \cdot c)$*

- **Identity element**: $\exists e \in G, \forall a \in G, a \cdot e = e \cdot a = a$.

- **Inverse element**: $\forall a \in G, \exists a^{-1} \in G, a \cdot a^{-1} = a^{-1} \cdot a = e$.

**Definition A.2.** *Let $G$ be a group and $X$ be an arbitrary set. A **left group action** is a map $\alpha : G \times X \to X$, that satisfies the following axioms:*

- $\alpha(e, x) = x$

- $\alpha(g, \alpha(h, x)) = \alpha(gh, x)$

*Note here we use the juxtaposition $gh$ to denote the binary operation in the group. If we shorten $\alpha(g, x)$ by $g \cdot x$, it's equivalent to say that $e \cdot x = x$, and $g \cdot (h \cdot x) = (gh) \cdot x$*

Note each $g \in G$ induces a map $L_g : X \to X, x \mapsto g \cdot x$.

### A.4.2 Homogeneous Riemannian manifolds, symmetric spaces and spaces of constant curvature

Let $\mathcal{M}$ be a Riemannian manifold and $I(\mathcal{M})$ be the set of all isometries of $\mathcal{M}$, that is, given $g \in I(\mathcal{M}), d(g \cdot x, g \cdot y) = d(x, y)$, for all $x, y \in \mathcal{M}$. It is clear that $I(\mathcal{M})$ forms a group, and thus, for a given $g \in I(\mathcal{M})$ and $x \in \mathcal{M}, g \cdot x \mapsto y$, for some $y \in \mathcal{M}$ is a group action. We call $I(\mathcal{M})$ the isometry group of $\mathcal{M}$.

Consider $o \in \mathcal{M}$, and let $H = \text{Stab}(o) = \{h \in G \mid h \cdot o = o\}$, that is, $H$ is the **Stabilizer** of $o \in \mathcal{M}$. Given $g \in I(M)$, its linear representation $g \mapsto d_x g$ in the tangent space $T_x \mathcal{M}$ is called the **isotropy representation** and the linear group $d_x \text{Stab}(x)$ is called the **isotropy group** at the point $x$.

We say that $G$ acts transitively on $\mathcal{M}$, iff, given $x, y \in \mathcal{M}$, there exists a $g \in \mathcal{M}$ such that $y = g \cdot x$.

**Definition A.3** ([Helgason, 1962])**.** *Let $G = I(\mathcal{M})$ act transitively on $\mathcal{M}$ and $H = \text{Stab}(o), o \in \mathcal{M}$ (called the "origin" of $\mathcal{M}$ ) be a subgroup of $G$. Then $\mathcal{M}$ is called a **homogeneous Riemannian manifold** and can be identified with the quotient space $G/H$ under the diffeomorphic mapping $gH \mapsto g \cdot o, g \in G$.*

By definition, we have $d(x, y) = d(g \cdot x, g \cdot y)$ for any $g \in G$ and any $x, y \in \mathcal{M}$. More importantly, any integrable function $f : \mathcal{M} \to \mathbb{R}$, we have [Helgason, 1962]

$$\int_{\mathcal{M}} f(x) d\nu(x) = \int_{\mathcal{M}} f(g \cdot x) d\nu(x)$$

This property leads to Proposition 4.1.

**Definition A.4** ([Helgason, 1962])**.** *A Riemannian symmetric space is a Riemannian manifold $\mathcal{M}$ such that for any $x \in \mathcal{M}$, there exists $s_x \in G = I(\mathcal{M})$ such that $s_x \cdot x = x$ and $ds_x|_x = -I$. $S_x$ is called symmetry at $x$.*

That is, a Riemannian symmetric space is a Riemannian manifold $\mathcal{M}$ with the property that the geodesic reflection at any point is an isometry of $\mathcal{M}$. Note that any Riemannian symmetric space is a homogeneous Riemannian manifold, but the converse is not true.

**Definition A.5** ([Vinberg et al., 1993])**.** *A simply-connected homogeneous Riemannian manifold is said to be a **space of constant curvature** if its isotropy group (at each point) is the group of all orthogonal transformations with respect to some Euclidean metric.*

Once again, a space of constant curvature is a symmetric space but the converse is not true.

### A.5 Theorem 4.1

*Proof.* Let $G$ be the isometry group of $\mathcal{M}$. Let $\eta_1, \eta_2 \in \mathcal{M}$ be arbitrary points such that $d(\eta_1, \eta_2) = \Delta$. By Corollary 4.1, the set $A$ reduces to $A = \left\{y \in \mathcal{M} : d(\eta_2, y)^2 - d(\eta_1, y)^2 \geq 2\sigma^2 \varepsilon\right\}$.

What we need to show is the following, for any points $\eta_1', \eta_2' \in \mathcal{M}$ such that $d(\eta_1', \eta_2') = \Delta$,

$$\int_A p_{\eta_1, \sigma}(y) \, d\nu(y) - e^\varepsilon \int_A p_{\eta_2, \sigma}(y) \, d\nu(y) = \int_{A'} p_{\eta_1', \sigma}(y) \, d\nu(y) - e^\varepsilon \int_{A'} p_{\eta_2', \sigma}(y) \, d\nu(y)$$

where $A' = \{y \in \mathcal{M} : d(\eta_2', y)^2 - d(\eta_1', y)^2 \geq 2\sigma^2 \varepsilon\}$. It's sufficient to show

$$\int_A p_{\eta_1,\sigma}(y) = \int_{A'} p_{\eta_1',\sigma}(y) \, d\nu(y), \quad \int_A p_{\eta_2,\sigma}(y) = \int_{A'} p_{\eta_2',\sigma}(y) \, d\nu(y). \qquad (14)$$

We can separate the proof into three cases: (1) $\eta_1' = \eta_1, \eta_2' \neq \eta_2$; (2) $\eta_1' \neq \eta_1, \eta_2' = \eta_2$; (3) $\eta_1' \neq \eta_1, \eta_2' \neq \eta_2$.

Case (1): $\eta_1' = \eta_1, \eta_2' \neq \eta_2$:

It follows that $\eta_2$ is in the sphere centered at $\eta$ with radius $\Delta$. (14) then follows from the rotational symmetry of the constant curvature spaces.

Case (2): $\eta_1' \neq \eta_1, \eta_2' = \eta_2$:

Same as case (1).

Case (3): $\eta_1' \neq \eta_1, \eta_2' \neq \eta_2$:

For any $\eta_1' \neq \eta_1$, there exists $g \in G$, such that $g \cdot \eta_1 = \eta_1'$. Denote $\eta_2' = g \cdot \eta_2$, we have

$$\begin{aligned}
gA :=&\{g \cdot y : d(\eta_2, y)^2 - d(\eta_1, y)^2 \geq 2\sigma^2 \varepsilon\} \\
=&\{g \cdot y : d(\eta_2', g \cdot y)^2 - d(\eta_1', g \cdot y)^2 \geq 2\sigma^2 \varepsilon\} \\
=&\{y : d(\eta_2', y)^2 - d(\eta_1', y)^2 \geq 2\sigma^2 \varepsilon\} \\
=&A'.
\end{aligned}$$

Let $F(y) := p_{\eta_1,\sigma}(y)\mathbf{1}_A(y)$, we have

$$\begin{aligned}
&\int_A p_{\eta_1,\sigma}(y) \, d\nu(y) \\
=&\int_{\mathcal{M}} F \circ L_g^{-1}(y) \, d\nu(y) \\
=&\int_{\mathcal{M}} p_{\eta_1,\sigma}(g^{-1} \cdot y)\mathbf{1}_{gA}(y) \, d(L_g^{-1})^* \nu(y); \quad \text{change of variable formular,} \\
=&\int_{\mathcal{M}} p_{\eta_1,\sigma}(g^{-1} \cdot y)\mathbf{1}_{gA}(y) \, d\nu(y); \quad \nu \text{ is a } G\text{-invariant measure,} \\
=&\frac{1}{Z_\sigma} \int_{gA} e^{-\frac{1}{2\sigma^2} d(g^{-1} \cdot y, \eta_1)^2} \, d\nu(y) \\
=&\frac{1}{Z_\sigma} \int_{gA} e^{-\frac{1}{2\sigma^2} d((gg^{-1}) \cdot y, g \cdot \eta_1)^2} \, d\nu(y) \\
=&\frac{1}{Z_\sigma} \int_{gA} e^{-\frac{1}{2\sigma^2} d(y, \eta_1')^2} \, d\nu(y) \\
=&\int_{gA} p_{\eta_1',\sigma(y)} d\nu(y) \\
=&\int_{A'} p_{\eta_1',\sigma(y)} d\nu(y).
\end{aligned}$$

For $\int_A p_{\eta_2,\sigma}(y) d\nu(y)$, the proof is the same. Combine with the result of case (1), we have finished the proof for case (3).

$\square$

## A.6 Simulation Details

For sampling from Riemannian Gaussian distribution $N_{\mathcal{M}}(\theta, \sigma^2)$ on $S^1$ (section 4.2), we first sample from truncated normal distribution with $\mu = 0$ and $\sigma^2$, then embed the sample to $\mathbb{R}$ and lastly counter-wise rotate the sample with degree $\theta$.

For simulation on $S^2$ (section 5.2), we choose the pair of $\eta$ and $\eta'$ to be $(1, 0, 0)$ and $(\cos(\Delta), (1 - \cos(\Delta)^2)^{1/2}, 0)$. Though any pair $\eta, \eta' \in S^2$ with $d(\eta, \eta') = \Delta$ works, we simply choose this specific pair for convenience. For Fréchet mean computation, we use a gradient descent algorithm described in Reimherr and Awan [2019].

### A.6.1  R Codes

For simulations in section 4.2, refer to R files euclid_functions.R & euclid_simulation.R for Euclidean space and sphere_functions.R & s1_simulation.R for unit circle $S^1$. For simulations in section 5.2, refer to R files sphere_functions.R & sphere_simulation.R.

