# OpenReview forum: "Gaussian Differential Privacy on Riemannian Manifolds"
_NeurIPS.cc/2023/Conference — NeurIPS 2023 poster_

### Official Review · Reviewer_vrR5 · 2023-06-30

**Soundness:** 3 good
**Presentation:** 3 good
**Contribution:** 3 good
**Rating:** 7
**Confidence:** 4

**Summary:**

The paper extends definition of Gaussian DP to General Riemannian Manifolds and proposed Riemannian Gaussian mechanism that shows to achieve GaussianDP. Paper also develops MCMC procedure for Riemannian Manifolds with constant curvature.


**Strengths:**

* Extending GaussianDP to Riemannian Manifolds
* MCMC procedure for Riemannian manifolds of constant curvature.

**Weaknesses:**

The experimental setup is extremely limited.

* **Baselines** The experiment compares the $\epsilon$-differential privacy mechanism (Riemannian Laplace mechanism) with the $(\epsilon, \delta)$-differential privacy mechanism (GDP mechanism), but it fails to include baselines that use $(\epsilon, \delta)$-mechanisms. It is necessary to include comparison with the following:

     *  DP-Riemannian Optimization ($\epsilon, \delta$)-differential privacy mechanism (source: https://arxiv.org/abs/2205.09494)

     Additionally, it is recommended to compare with
    *  Riemannian K-Norm Gradient mechanism ($\epsilon$-DP) (source: https://arxiv.org/pdf/2209.12667.pdf).

*  **Experiments used only one manifold and of max dimension 3 for n=10 samples:** Experiments were solely carried out on 2-dimensional ($S^1$) and 3-dimensional ($S^2$) manifolds. It is essential to extend the experimentation to include at least one additional manifold, such as a hyperbolic manifold, which represents another constant curvature manifold. Furthermore, it is advisable to introduce variations in the parameters $n$ and $d$ during the experiments.

Related work :
Also related work on DP-Optimization Improved Differentially Private Riemannian Optimization: Fast Sampling and Variance Reduction , TMLR 2023, (https://openreview.net/pdf?id=paguBNtqiO)

**Questions:**

NA

---

> ### Author Rebuttal · Authors · 2023-08-07
>
> Dear Reviewer,
>
> Thank you for your thorough review and constructive feedback. We appreciate your insights, and we have addressed your concerns and questions in our response below:
>
> ## Weakness
>
> > The experiment compares the $\varepsilon$-differential privacy mechanism (Riemannian Laplace mechanism) with the $(\varepsilon, \delta)$-differential privacy mechanism (GDP mechanism), but it fails to include baselines that use -mechanisms. It is necessary to include comparison with the following: DP-Riemannian Optimization $(\varepsilon, \delta)$-differential privacy mechanism
>
> The reason we are comparing to the Riemannian Laplace mechanism that satisfies $\varepsilon$-DP is due to the fact any mechanism that achieves $\varepsilon$-DP can achieve $\mu$-GDP for some privacy budget $\mu$ [Liu et al., 2022].  Therefore, our intention is to compare two mechanisms that both can achieve GDP. However, the same thing cannot be said for $(\varepsilon, \delta)$-DP mechanism. More specifically,  a mechanism that achieves $(\varepsilon, \delta)$-DP cannot guarantee $\mu$-GDP for any privacy budget $\mu$.
>
> > Additionally, it is recommended to compare with Riemannian K-Norm Gradient mechanism ($\varepsilon$-DP)
>
> We have considered the k-norm mechanism as part of the comparison but quickly dismissed it due to its uses of a different approach for sensitivity computation, so it could be an unfair comparison. However, on second thought, it could provide some insight regardless of its sensitivity computation. We will include the $k$-norm mechanism as a comparison for future works.
>
> >  It is essential to extend the experimentation to include at least one additional manifold, such as a hyperbolic manifold, which represents another constant curvature manifold.
>
> Yes, we do agree on more manifolds are needed. We only experiment on the sphere due to the sampling methods on it being well established. We will work to extend to more manifolds in the future.
>
> > Furthermore, it is advisable to introduce variations in the parameters $n$ and $d$ during the experiments.
>
> The parameter $n$ will only affect the sensitivity calculation. A large $n$ will result in small sensitivity, which requires less noise perturbation in general. We choose a small $n$ to better illustrate the improvement in utility.
>
> > Related work: Also related work on DP-Optimization Improved Differentially Private Riemannian Optimization: Fast Sampling and Variance Reduction, TMLR 2023
>
> Thank you for pointing it out. It will be added in the revision.

---

> > ### Comment · Reviewer_vrR5 · 2023-08-10
> > **Thanks for rebuttal.**
> >
> > Thanks for rebuttal. I am happy with rebuttal. I will increase my score

---

> > > ### Author Response · Authors · 2023-08-14
> > >
> > > We appreciate your positive feedback! We'll make sure to incorporate your other constructive suggestions in our future work.

---

### Official Review · Reviewer_3t86 · 2023-06-30

**Soundness:** 3 good
**Presentation:** 3 good
**Contribution:** 3 good
**Rating:** 7
**Confidence:** 4

**Summary:**

The authors consider the problem of achieving Gaussian Differential Privacy (GDP) on Riemannian manifolds.  After defining what this concept means, they provide a mechanism capable of achieving GDP via a generalization of the Gaussian to manifolds.  The major challenge in implementing the approach is in finding a suitable scaling parameter, $\sigma$, for the noise.  The authors define several equations that must be solved to find $\sigma$, each which become more tractable as additional structural assumptions on the manifold are made.

**Strengths:**

This is a timely paper as there has been an increased interest in manifolds in statistics/machine learning.  There have also been several recent works that consider privacy within these contexts as well.  The Gaussian mechanism is one of the most popular tools for achieving privacy, so it is quite natural to consider it in the context of manifolds.  The extension of the Gaussian mechanism to manifolds is much more challenging than the Laplace.

**Weaknesses:**

Implementing the mechanism seems quite challenging.  Just to find the scaling parameter $\sigma$ is difficult in general, plus one must do something like MCMC to actually sample from the mechanism.  This limits the practical impact of the paper.

**Questions:**

Def 3.1: I would emphasize that this is a known equivalence from the Dong et al paper.  I think the authors should also explain a bit more why this definition is easier to work with on manifolds than the original gaussian dp.  In general, I’m not convinced that it truly is.  The GDP definition is simply a statement about the tradeoff between type 1 and type 2 errors, it really has nothing to do with the underlying space.  So the central challenge isn’t defining GDP for manifolds, but defining a mechanism that achieves it.

“In contrast to the task of releasing manifold-valued private summary, Han et al. [2022] considers a different scenario where the private summary resides on the tangent bundle of Riemannian manifolds and extends (ε, δ)-DP and its Gaussian mechanism to general manifolds."
This seems a bit inaccurate since their intention was not to privately release the gradient but rather use the private gradient for the private summary which resides on the manifold.  I think the wording here should be clarified.

Minor— I don’t think the authors ever actually define $\Phi$.

---

> ### Author Rebuttal · Authors · 2023-08-07
>
> Dear Reviewer,
>
> Thank you for your thorough review and constructive feedback. We appreciate your insights, and we have addressed your concerns and questions in our response below:
>
> ## Weakness
> > Implementing the mechanism seems quite challenging. Just to find the scaling parameter is difficult in general, plus one must do something like MCMC to actually sample from the mechanism. This limits the practical impact of the paper.
>
> Yes, this is one of the difficulties when working under the general manifold setting. The utility gains from utilizing the intrinsic geometry of the data manifold come with the cost of computational complexity. It really depends on whether the end user is willing to make the trade-off for better utility, and hopefully, more end users will be willing to make the trade-off as computing technology advances.
>
> ## Questions
>
> > Def 3.1: I would emphasize that this is a known equivalence from the Dong et al paper.
>
> Thank you for the suggestion. The emphasis will be added in the revision.
>
> > I think the authors should also explain a bit more why this definition is easier to work with on manifolds than the original gaussian dp. In general, I’m not convinced that it truly is. The GDP definition is simply a statement about the tradeoff between type 1 and type 2 errors, it really has nothing to do with the underlying space.
>
> I disagree partially here. In order to extend GDP to general manifolds using the hypothesis interpretation, the hypothesis test should reflect the underlying geometry. More specifically, in Euclidean spaces, $\mu$-GDP relates to the hypothesis testing problem of distinguishing between $N(0, 1)$ and $N(\mu, 1)$. If we were to extend to general manifolds, we should consider the hypothesis testing of distinguishing between Riemannian manifold generalization of Gaussian distribution. However, there are different ways of generalizing Gaussian distribution as mentioned briefly after Definition 3.3. Regardless of which extension to use, it is difficult to determine the optimal trade-off between type I and type II errors under this Riemannian manifold hypothesis testing setting. For instance, computing the ratio between two likelihoods for the likelihood ratio test is a difficult task.
>
> > So the central challenge isn’t defining GDP for manifolds, but defining a mechanism that achieves it.
>
> Yes, the challenge lies in designing a mechanism that achieves GDP while utilizing as much of the privacy budget as possible (That is, we want the left-hand side of equation (2) to be as close to the right-hand side as possible).
>
> > This seems a bit inaccurate since their intention was not to privately release the gradient but rather use the private gradient for the private summary which resides on the manifold. I think the wording here should be clarified.
>
> The wording regarding Han et al. [2022] will be adjusted to clarify that their focus is to solve empirical risk minimization problems in a $(\varepsilon, \delta)$-DP compliant manner by privatizing the gradient which resides on the tangent bundle of Riemannian manifolds.
>
> > Minor— I don’t think the authors ever actually define $\Phi$.
>
> Thank you for pointing it out. It accidentally got deleted during the adjustment process. It will be added in the revision.

---

> > ### Comment · Reviewer_3t86 · 2023-08-18
> >
> > Thank you for the response.  I get your point about the hypothesis testing and the underlying geometry.  I would acknowledge that it is at least a little ambiguous and your approach makes the application to manifolds cleaner, but it isn't obvious that the underlying space needs to be considered when defining the type of space (the underlying space plays no role in the privacy definition for pure DP, approximate DP, etc).  The f-DP framework just says that the function f is tradeoff between type 1 and type 2 errors, and those errors could come from anywhere.  Regardless, I still consider this a minor (though interesting) point.

---

> > > ### Author Response · Authors · 2023-08-19
> > >
> > > Thanks for the positive feedback. We understand that it isn't obvious that the underlying space needs to be considered when considering the hypothesis testing approach. To make the use of Definition 3.1 more natural and obvious, we will add more explanations in the revision. For convenience, we will add more explanations here as well to address any lingering concerns you might still hold.
> > >
> > > Consider a query result that resides on the unit sphere $S^2$. To privatize it in a GDP-compliant way, we have two options outlined below.
> > > 1. Commonly referred to as the extrinsic approach, we can simply disregard the unit sphere $S^2$ and consider the query result lies on $\mathbb{R}^3$ instead. The rest is trivial as we can apply the existing framework of GDP on Euclidean space. However, the major shortcoming of this approach is the absence of assurance that the privatized query will still reside on $S^2$.
> > > 2. To address this issue, we need to take $S^2$ into consideration. This is referred to as the intrinsic approach. To guarantee that a GDP-compliant mechanism $M$ will produce results within $S^2$, we need to ensure the distribution of its outcome is defined entirely on $S^2$. Therefore, the hypothesis testing approach relates to the problem of distinguishing between $M(\mathcal{D})$ and $M(\mathcal{D}')$, two distributions on $S^2$ instead of $\mathbb{R}^3$. To determine the optimal type 1 and type 2 trade-off $T(M(\mathcal{D}), M(\mathcal{D}'))$ between these distributions is difficult. For instance, computing the ratio between two likelihoods can be quite involved due to the complexity of the normalizing constant (e.g., the dependence of $Z(\eta, \sigma)$ on $\eta$). This is where Definition 3.1 comes in, using the connection with $(\varepsilon, \delta)$-DP, we can establish GDP on $S^2$ purely based on the notion of Borel probability measure.

---

### Official Review · Reviewer_12f9 · 2023-07-04

**Soundness:** 2 fair
**Presentation:** 3 good
**Contribution:** 2 fair
**Rating:** 6
**Confidence:** 3

**Summary:**

To obtain manifold-value statistic with the Gaussian differential privacy (GDP), the authors extend the GDP from Euclidean spaces to general Riemannian manifolds. Then they proved that their proposed Riemannian Gaussian distribution make GDP tractable. Furthermore, they proposed an efficient numerical algorithm to compute the privacy budget μ on the manifolds of one dimension, especially on the manifolds with constant curvature. At last, they provided some simulation experiments to illustrate performance of their algorithms.

Due to the importance of GDP among the variants of DP, it is meaningful to generalize GDP to Riemannian manifolds. They solved this question partially. Considering the importance of this question, the paper is worth to be accepted. Also, this paper is organized clearly except some typos. I have checked carefully some important details of proofs, so the results should be reliable.

**Strengths:**

1.	They research the extension of an important variant of DP—GDP and prove that they can achieve GDP on Riemannian manifolds by using Riemannian Gaussian distribution.
2.	Following previous works, their results make sense.


**Weaknesses:**

Some typos influence the readability of this paper. For example, ‘compliant’, not ‘compliment’ in subsection Main Contributions; and $N_m(\eta,\sigma^2)$  is correct, not $N_m(\mu,\sigma^2)$.
2. For the case of manifolds of dimension $d>1$, they make a too strong assumption that the manifold with constant curvature to design an efficient numerical algorithm. To the best of my knowledge, this assumption can’t encompass enough manifolds (I can only think of Euclidean space, the sphere and Lobachevskij space). The motivation of this assumption is not strong in practice.


**Questions:**

1.	This paper mentioned all hypothesis testing-based privacy definitions converge to the guarantees of GDP in the limit of composition and when the dimension of the privatized data approaches infinity, a large class of noise addition private mechanisms is shown to be asymptotically GDP. Are these results valid on Riemannian manifolds?
2. 	It is hard to solving the case of manifolds of dimension $d>1$, which make us use numerical methods to compute the integral. This work uses MCMC to approximate the integral, how about other numerical integration methods?
  3.	They claimed that their work is the extension of an important variant of DP, so this work may be valuable in practice. Thus they should explain the point clearly in the introduction and provide some real world numerical examples.

**Limitations:**

See "weakness".

---

> ### Author Rebuttal · Authors · 2023-08-07
>
> Dear Reviewer,
>
> Thank you for your thorough review and constructive feedback. We appreciate your insights, and we have addressed your concerns and questions in our response below:
>
> ## Weakness
>
> >  For the case of manifolds of dimension $d>1$, they make a too strong assumption that the manifold with constant curvature to design an efficient numerical algorithm. To the best of my knowledge, this assumption can’t encompass enough manifolds (I can only think of Euclidean space, the sphere and Lobachevskij space). The motivation of this assumption is not strong in practice
>
> We agree the assumption of constant curvature space is restrictive. However, it serves as a necessary stepping stone for future works where we plan to relax the assumption to symmetric space/homogeneous space, which includes many frequently used manifolds like SPD space.
>
> ## Questions
>
> > This paper mentioned all hypothesis testing-based privacy definitions converge to the guarantees of GDP in the limit of composition and when the dimension of the privatized data approaches infinity, a large class of noise addition private mechanisms is shown to be asymptotically GDP. Are these results valid on Riemannian manifolds?
>
> The convergence to GDP on general manifolds is still an open question. Hopefully, we can address it in future works.
>
> > It is hard to solving the case of manifolds of dimension $d>1$, which make us use numerical methods to compute the integral. This work uses MCMC to approximate the integral, how about other numerical integration methods?
>
> We haven't considered other integration methods. The MCMC method is a natural method to use here due to the appearance of the probability density $p_{\eta, \sigma}$ inside the integral. However, the Bayesian quadrature method could be useful here as it can quantify the uncertainty of the integral estimation.
>
> > They claimed that their work is the extension of an important variant of DP, so this work may be valuable in practice. Thus they should explain the point clearly in the introduction and provide some real-world numerical examples.
>
> One of the practical importance of extending GDP to general manifolds is the better data utility it can achieve under some use cases (query composition and subsampling). The most notable example of nonlinear private data is in medical imaging. They reside in the space of SPD matrices. Although GDP can be extended to SPD space without issues, calibrating $\sigma$ efficiently requires the assumption of constant curvature. Hopefully, we can relax the assumption of symmetric space, which includes SPD space, in the future.

---

> > ### Comment · Reviewer_12f9 · 2023-08-14
> >
> > Thanks to your reply. Despite answers have been addressed, the weakness on the constant curvature assumption remains, and this reduces the technical difficulties and interests to applications.

---

> > > ### Author Response · Authors · 2023-08-14
> > >
> > > We appreciate your prompt response. While acknowledging that our approach relies on the constant curvature assumption, we believe it is a substantial advancement over prior research as we explain next.
> > >
> > > Previous studies have typically followed the following two approaches:
> > > 1. Focus on specific manifold and metric: For instance, in the work by [1], emphasis was solely placed on the SPD space utilizing a log-Euclidean metric.
> > > 2. Dependence on tangent spaces: As illustrated in [2], the approach revolved around tangent spaces and their utilization in resolving private empirical risk minimization issues through private gradient implementation.
> > >
> > > In contrast, our research improves upon both of the aforementioned aspects:
> > > 1. Our method is more general as the assumption of constant curvature encompasses more than one particular manifold.
> > > 2. Our method works directly on the manifold itself without relying on computation on the tangent spaces, and thus it's more flexible since we do not need to restrict to solve problems through the use of private gradient.
> > >
> > > [1] S. Utpala, P. Vepakomma, and N. Miolane. Differentially Private Fréchet Mean on the Manifold of Symmetric Positive Definite (SPD) Matrices with log-Euclidean Metric. Transactions on Machine Learning Research, 2023.
> > >
> > > [2] A. Han, B. Mishra, P. Jawanpuria, and J. Gao. Differentially Private Riemannian Optimization. arXiv preprint arXiv:2205.09494, 2022.

---

### Official Review · Reviewer_j3sD · 2023-07-05

**Soundness:** 1 poor
**Presentation:** 3 good
**Contribution:** 1 poor
**Rating:** 4
**Confidence:** 4

**Summary:**

This work studies an extension of a Gaussian privacy mechanism on Riemannian manifolds, so as to extend the notion of mu-GDP for estimators with given global sensitivity. The idea is to sample a random point on the manifold with a Gibbs density (wrt to the Riemannian volume) whose potential is proportional to the squared distance to the given estimator (that would simply be a Gaussian density on a Euclidean space). It is well known that in order for such a distribution to be well defined, it is enough to assume that the sectional (or, equivalenty, Ricci) curvature of the manifold is bounded from below. The main challenges, then, reside in : (1) tuning the temperature parameter of the Gibbs distribution so as to obtain mu-GDP, and (2) sample from such a distribution.


While giving general formulae for the choice of the temperature parameter, the authors study the case of homogeneous manifolds, as well as model manifolds with constant curvature, also providing simulation studies.

Differential privacy on non-linear spaces, in particular Riemannian manifolds, has not been studied a lot yet, and this paper aims at providing new, simple insights.

**Strengths:**

•	The question tackled in this paper is important, given the increasing need of dealing with non-linearity in data.
•	The paper is written clearly and the main text is easy to follow.

**Weaknesses:**

•	While the paper addresses an important question, the execution remains shallow. Laplace and Gaussian mechanisms can be very easily extended to Riemannian manifolds, and the computations that yield to determine the calibration of their parameters are very similar to the Euclidean case. One challenge that is barely addressed here (see the very brief remark just before Definition 4.1) is the understanding of the connexion between these parameters and the geometry (in particular, curvature bounds) of the ambient manifold. Instead, the authors choose to focus on very simple (and perhaps not realistic in most applications) manifolds, where everything becomes much simpler. Hence, I think that a big, if not essential, part of the picture is missing in this work.
•	The presence of Definition 3.1 (mu-GDP mechanisms on manifolds) is obscure to me: A definition is already given in Definition 2.2, and I do not see why it should be different on manifolds. Should Definition 3.1 actually be a theorem? I mean that one should check that M is mu-GDP in the sense of Definition 2.2 if and only if it is (eps,delta(eps))-DP for all eps>0, with delta(eps) given by the formula that appears in Definition 3.1 (something similar to Theorem A.1).
•	Some proofs seem either wrong or incomplete to me, here are some pointers:
		- One line after Theorem A.1, this theorem is interpreted using the O(eps^2) notation, which completely forgets the dependence in mu. Hence, the whole proof of Theorem 3.1 forgets mu and is, therefore, imprecise/incomplete.
		- I did not understand the first line of the proof of Theorem 3.1. Also, there, what is x (it does not appear in the formula)?
		- In Equation 4, I think that the normalizing constant Z_{eta,sigma} is missing in the upper bound.


Some minor remarks:
•	Line 92, there is a typo.
•	Line 99: « the » should be replaced with « a ».
•	Line 237: Remove « satisfies »
•	Every citation should come to a pointer to a specific definition or result, especially when the citation refers to a whole book (e.g., Dudley, 2002)!
•	In Theorem A.1, what is the ‘t’ in the subscript of mu_t?
•	Also in Theorem A.1, « privacy profile » has not be defined anywhere.



**Questions:**

See paragraph on weaknesses.

---

> ### Author Rebuttal · Authors · 2023-08-07
>
>
> Dear Reviewer,
>
> Thank you for your thorough review and constructive feedback. We appreciate your insights, and we have addressed your concerns and questions in our response below:
>
> ## Weakness
> > Laplace and Gaussian mechanisms can be very easily extended to Riemannian manifolds, and the computations that yield to determine the calibration of their parameters are very similar to the Euclidean case.
>
> We have to disagree here. We agree that for the Laplace mechanism, the extension to Riemannian manifolds is straightforward as demonstrated in Reimherr et al. [2021]. However, for the Gaussian mechanism, it's difficult to calibrate the parameter due to the lack of inner product structure under the general manifold setting. The existing literature either focuses on a specific manifold with zero curvature or on tangent spaces instead. We improve on the existing literature by providing a way to extend the Gaussian mechanism to general manifolds for GDP in Theorem 3.2 and designing an easy-to-implement algorithm to calibrate the noise parameter $\sigma$ in the case of constant curvature spaces.
>
> > One challenge that is barely addressed here (see the very brief remark just before Definition 4.1) is the understanding of the connexion between these parameters and the geometry (in particular, curvature bounds) of the ambient manifold.
>
> We agree that this is indeed an important challenge to tackle. It is partially addressed as we establish the connection between the privacy parameter $\mu$ and the underlying geometry implicitly by connecting $\mu$ with the noise parameter $\sigma$ of Riemannian Gaussian distribution in Theorem 3.2.
>
> > Instead, the authors choose to focus on very simple manifolds, where everything becomes much simpler. Hence, I think that a big, if not essential, part of the picture is missing in this work.
>
> As mentioned previously, the calibration of $\sigma$ is a difficult task without restricting a specific manifold with zero curvature or working on tangent space. Compared to previous works, the assumption of constant curvature is a significant improvement despite being simple. Additionally, it serves as a necessary stepping stone for future works where we plan to relax the assumption to symmetric space/homogeneous space, which includes many frequently used manifolds like SPD space.
>
> ### Regarding the use of definition 3.1:
> > The presence of Definition 3.1 (mu-GDP mechanisms on manifolds) is obscure to me.
>
> The original definition of GDP given in 2.2 is based on the hypothesis testing interpretation. If we were to extend to general manifolds, we should consider the hypothesis testing of distinguishing between Riemannian manifold generalization of Gaussian distribution. It would be a challenging task as obtaining the general manifold version of the optimal trade-off function $T(N(0,1), N(\mu, 1))$ would be difficult due to the following reasons:
> 1. There is no canonical way of extending Gaussian distribution to the general manifold. Different methods of extension (mentioned briefly after Definition 3.3) could lead to slightly different versions of GDP on manifolds.
> 2. Common ways of constructing hypothesis testing are difficult to implement. For instance, computing the ratio between two likelihoods for the likelihood ratio test is a difficult task.
> To bypass such difficulty, we introduce definition 3.1.
>
> > Should Definition 3.1 actually be a theorem?
>
> Definition 3.1 is based on a known equivalence between GDP and approximate DP on Euclidean spaces. Since approximate DP is well defined on the general manifold, this connection provides us with a natural way of extending the notion of GDP to the general manifold.
>
> ### Regarding the proof of Theorem 3.1:
> > In Theorem A.1, what is the $t$ in the subscript of $\mu_t$?
>
> The $t$ stands for "tail", we realized that the use of subscript $t$ is not appropriate here and causes confusion, and $\mu_{tail}$ will be used instead in the revision.
>
> > privacy profile has not been defined anywhere.
>
> We will add the definition of privacy profile in the revision. The privacy profile is a function associating to each privacy parameter $\alpha = e^{\varepsilon}$ bound on the $\alpha$-divergence between the results of running the mechanism on two adjacent datasets. Note that for our Riemannian Gaussian mechanism, this is exactly the left-hand side of equation (2).
>
> > One line after Theorem A.1, this theorem is interpreted using the $O(\varepsilon^2)$ notation, which completely forgets the dependence in $\mu$. Hence, the whole proof of Theorem 3.1 forgets $\mu$ and is, therefore, imprecise/incomplete
>
> Theorem A.1 implies a privacy mechanism with finite $\mu_{tail}$ can achieve $\mu$-GDP for some $\mu$. The exact value of $\mu$ is not the focus here as Theorem 3.1 simply says that the Riemannian Gaussian mechanism can achieve $\mu$-GDP for some $\mu > 0$. To determine the exact value of $\mu$, we refer to Theorem 3.2 and Theorem 4.1. We will add more explanations in the appendix for better clarity.
>
> > Also, there, what is $x$ (it does not appear in the formula)?
>
> This is a typo, it should be $\eta$ instead.
>
> > I did not understand the first line of the proof of Theorem 3.1.
>
> The key idea is to show $\mu_{tail}$ is finite, which is implied if $-\log(\delta_{\mathcal{A}}(\varepsilon))$ is $O(\varepsilon^{-2})$. Since the privacy profile $\delta_{\mathcal{A}}(\varepsilon)$ can be represented by the left-hand side of equation (2), we have the first line of the proof.
>
> > In Equation 4, I think that the normalizing constant is missing in the upper bound.
>
> Yes, it will be corrected in the revision. However, it will not affect the rest of the proof since it is finite and does not depend on $\varepsilon$.
>
> > Every citation should come to a pointer to a specific definition or result.
>
> Thank you for the suggestion. The citations will be adjusted to include specific definitions/results for better readability.

---

> > ### Comment · Reviewer_j3sD · 2023-08-16
> >
> > Thank you for the detailed rebuttal.
> >
> > I am willing to increase my score from 3 to 4, but I still believe that this work misses precise quantitative connections between the privacy mechanism and the geometry of the space, which is a major limitation, in my opinion.

---

> > > ### Author Response · Authors · 2023-08-16
> > >
> > > We greatly appreciate your decision to increase the score. While we fully recognize the significance of establishing precise quantitative connections between the privacy mechanism and the geometry of the space, addressing this challenging task deserves more research efforts in future works. To the best of our knowledge, no previous work has attempted to tackle this fundamental problem. Nevertheless, our novel framework paved the way for future investigation.
> > >
> > > In particular, the first essential step is to extend GDP (approximate DP) and its privacy mechanism to general manifolds. However, previous works have focused on specific manifolds, e.g., the work in [1], therefore making it impossible to study the connection between its geometry and the privacy mechanism. The reason is that we need to observe how the privacy mechanism responds to different geometries (e.g., a change in curvature) in order to make the connection. For example, if the privacy mechanism were extendable to both spherical and hyperbolic spaces, we could then examine how the calibrations of the noise parameter $\sigma$ differ between these two spaces.
> > >
> > > In contrast, we extend GDP to general manifolds (definition 3.1) and provide a way to extend the Gaussian mechanism to general manifolds (Theorem 3.2). Additionally, we design a simple algorithm to efficiently calibrate the noise parameter $\sigma$ for constant curvature spaces. This is an important first step. That is, once the privacy mechanism has been extended to a wide range of manifolds and its parameter $\sigma$ can be efficiently calibrated, we can then start further investigation of the connection. However, considering that it is beyond the scope of our current work, we intend to tackle this problem in future research.
> > >
> > > [1] S. Utpala, P. Vepakomma, and N. Miolane. Differentially Private Fréchet Mean on the Manifold of Symmetric Positive Definite (SPD) Matrices with log-Euclidean Metric. Transactions on Machine Learning Research, 2023.

---

### Author Response · Authors · 2023-08-21
**Dear Area Chair and Reviewers**

After the initial submission, we received many constructive and insightful feedback from the reviewers. We express our gratitude to the reviewers for acknowledging the importance of extending GDP and the Gaussian mechanism to general manifolds, even though it is primarily applicable to a specific subset of manifold structures. As we proceed, we will outline some of the key concerns that have arisen in the reviews.

1. One of the recurring concerns that reviewers is the use of Definition 3.1. We have addressed this by giving a concrete example in our reply to reviewer 3t86.
2. In our response, we have extensively clarified the aspects of confusion raised by Reviewer j3sD regarding the proof of Theorem 3.1, providing additional explanations and background results.

We extend our sincere gratitude for reviewing our paper. We value the engaging discourse and invaluable feedback you have provided.

---

### Decision · Program_Chairs · 2023-09-21

**Decision:**

Accept (poster)

**Comment:**

This paper effectively explores Gaussian Differential Privacy in the context of Riemannian manifolds, addressing a significant topic in the current machine learning landscape. While the theoretical contribution is solid and timely, practical implementation appears challenging, potentially restricting its immediate impact. The meta-reviewer suggests accepting this work as a poster to facilitate further refinement.